# VoiceFixer: Toward General Speech Restoration With Neural Vocoder

## Abstract

Speech restoration aims to remove distortions in speech signals. Prior methods mainly focus on *single-task speech restoration* (SSR), such as speech denoising or speech declipping. However, SSR systems do not address the general speech restoration problem and show limited performance in some speech restoration tasks such as speech super-resolution. To overcome those limitations, we propose a *general speech restoration* (GSR) task that attempts to remove multiple distortions simultaneously. Furthermore, we propose *VoiceFixer*[1], a generative framework to address the GSR task. *VoiceFixer* consists of an *analysis stage* and a *synthesis stage* to mimic the speech analysis and comprehension of the human auditory system. We employ a ResUNet to model the analysis stage and a neural vocoder to model the synthesis stage. We evaluate *VoiceFixer* with additive noise, room reverberation, low-resolution, and clipping distortions. Our baseline GSR model achieves a 0.499 higher mean opinion score (MOS) than the speech denoising SSR model. *VoiceFixer* further surpasses the GSR baseline model on the MOS score by 0.256. Moreover, we observe that *VoiceFixer* generalizes well to severely degraded real speech recordings, indicating its potential in restoring old movies and historical speeches.

## 1 Introduction

Speech restoration is a process to restore degraded speech signals to high-quality speech signals. Speech restoration is an important research topic since distortions are ubiquitous. For example, speech is usually surrounded by background noise, blurred by room reverberations, or recorded by low-quality devices (Godsill et al., 2002). Those distortions degrade the perceptual quality of speech for human listeners. Speech restoration has a wide range of applications such as online meeting (Defossez et al., 2020), hearing aids (Van den Bogaert et al., 2009), and audio editting (Van Winkle, 2008). Still, speech restoration remains a challenging problem due to the large variety of distortions in the world.

Previous works in speech restoration mainly focus on *single task speech restoration* (SSR), which deals with only one type of distortion at a time. For example, speech denoising (Loizou, 2007), speech dereveberation (Naylor & Gaubitch, 2010), speech super-resolution (Kuleshov et al., 2017), or speech declipping (Záviška et al., 2020). However, in the real world, speech signal can be degraded by several different distortions simultaneously, which means previous SSR systems oversimplify the speech distortion types (Kashani et al., 2019; Lin et al., 2021; Kuleshov et al., 2017; Birnbaum et al., 2019). The mismatch between the training data used in SSR and the testing data from the real world degrades the speech restoration performance.

To address the mismatch problem, we propose a new task called general speech restoration (GSR), which aims at restoring multiple distortions in a single model. A numerous studies (Cutler et al., 2021; Cauchi et al., 2014; Han et al., 2015) have reported the benefits of jointly training multiple speech restoration tasks. Nevertheless, performing GSR using one-stage systems still suffer from the problems in each SSR task. For example, on generative tasks such as speech super-resolution (Kuleshov et al., 2017), one-stage models tend to overfitting the filter (Sulun & Davies, 2020) used during training. Based on these observations, we propose a two-stage system called *VoiceFixer*.

---

[1]Restoration samples: `https://anonymous20211004.github.io/demo-vf/`

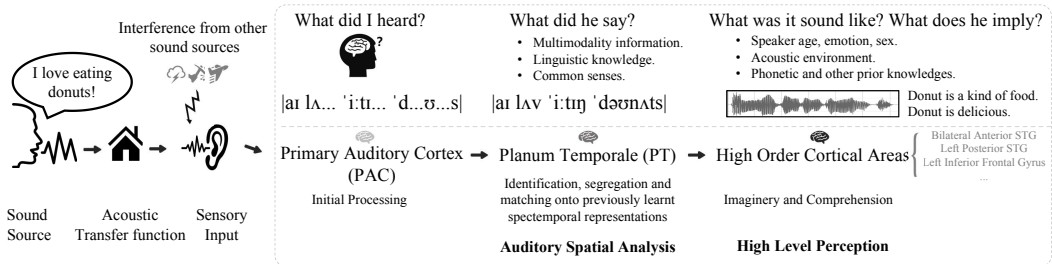

**Figure 1:** The neural and cognitive model of how human brain understand and restore distorted speech.

The design of *VoiceFixer* is motivated by the biological mechanisms of human hearing when restoring distorted speech. Intuitively, if a person tries to identify a strongly distorted voice, his/her brain can do the recovery by utilizing both the distorted speech signal and the prior knowledge of the language. As shown in Figure 1, the speech distortion perception is modeled by neuroscientists as a two-stage process, including an auditory scene analysis stage (Bregman, 1994), and a high level comprehension/synthesis stage (Griffiths & Warren, 2002). In the analysis stage, the sound information is first transformed into acoustic features by primary auditory cortex (PAC). Then planum temporale (PT), the cortical area posterior to the auditory cortex, acts as a computational hub by segregating and matching the acoustic features to low level spectrotemporal representations. In the synthesis stage, a high order cortical area is hypothesised to perform the high level perception tasks (Griffiths & Warren, 2002; Kennedy-Higgins, 2019). Our proposed *VoiceFixer* systems model the analysis stage with spectral transformations and a deep residual UNet, and the synthesis stage with a convolutional vocoder trained using adversarial losses. One advantage of the two-stage *VoiceFixer* is that the analysis and synthesis stages can be trained separately. Two-stage methods have also been successfully applied to the speech synthesis task (Wang et al., 2016; Ren et al., 2019) where acoustic models and vocoders are trained separately.

*VoiceFixer* is the first GSR model that is able to restore a wide range of low-resolution speech sampled from 2 kHz to 44.1 kHz, which is different from previous studies working on constant sampling rates (Lim et al., 2018; Wang & Wang, 2021; Lee & Han, 2021). To the best of our knowledge, *VoiceFixer* is the first model that jointly performs speech denoising, speech dereverberation, speech super-resolution, and speech declipping in a unified model.

The rest of this paper is organized as follows. Section 2 introduces the formulations of speech distortions. Section 3 describes the design of *VoiceFixer*. Section 4 discusses the evaluation results. Appendixes introduce related works and show speech restoration demos.

## 2 PROBLEM FORMULATIONS

We denote a segment of a speech signal as $s \in \mathbb{R}^L$, where $L$ is the samples number in the segment. We model the distortion process of the speech signal as a function $d(\cdot)$. The degraded speech $x \in \mathbb{R}^L$ can be written as:

$$x = d(s). \tag{1}$$

Speech restoration is a task to restore high-quality speech $\hat{s}$ from $x$:

$$\hat{s} = f(x) \tag{2}$$

where $f(\cdot)$ is the restoration function and can be viewed as a reverse process of $d(\cdot)$. The target is to estimate $s$ by restoring $\hat{s}$ from the observed speech $x$. Recently, several deep learning based one-stage methods have been proposed to model $f(\cdot)$ such as fully connected neural networks, recurrent neural networks, and convolutional neural networks. Detailed introductions can be found in Appendix A.

**Distortion modeling** is an important step to simulate distorted speech when building speech restoration systems. Several previous works model distortions in a sequential order (Vincent et al., 2017; Tan et al., 2020; Zhao et al., 2019). Similarly, we model the distortion $d(\cdot)$ as a composite function:

$$d(\boldsymbol{x}) = d_1 \circ d_2 \circ ... d_Q(\boldsymbol{x}), d_q \in \mathbb{D}, q = 1, 2, ..., Q, \tag{3}$$

where $\circ$ stands for function composition and $Q$ is the number of distortions to consist $d(\cdot)$. Set $\mathbb{D} = \{d_v(\cdot)\}_{v=1}^{V}$ is the set of distortion types where $V$ is the total number of types. Equation 3 describes the procedure of compounding different distortions from $\mathbb{D}$ in a sequential order. We introduce four speech distortions as follows.

**Additive noise** is one of the most common distortion and can be modeled by the addition between speech $\boldsymbol{s}$ and noise $\boldsymbol{n} \in \mathbb{R}^L$:

$$d_{\text{noise}}(\boldsymbol{s}) = \boldsymbol{s} + \boldsymbol{n}. \tag{4}$$

**Reverberation** is caused by the reflections of signal in a room. Reverberation makes speech signals sound distant and blurred. It can be modeled by convolving speech signals with a room impulse response filter (RIR) $\boldsymbol{r}$:

$$d_{\text{rev}}(\boldsymbol{s}) = \boldsymbol{s} * \boldsymbol{r}, \tag{5}$$

where $*$ stands for convolution operation.

**Low-resolution** distortions refer to audio recordings that are recorded in low sampling rates or with limited bandwidth. There are many causes for low-resolution distortions. For example, when microphones have low responses in high-frequency, or audio recordings are compressed to low sampling rates, the high frequencies information will be lost. We follow the description in Wang & Wang (2021) to produce low-resolution distortions but add more filter types (Sulun & Davies, 2020). After designing a low pass filter $\boldsymbol{h}$, we first convolve it with $\boldsymbol{s}$ to avoid the aliasing phenomenon. Then we perform resampling on the filtered result from the original sampling rate $o$ to a lower sampling rate $u$:

$$d_{\text{low\_res}}(\boldsymbol{s}) = \text{Resample}(\boldsymbol{s} * \boldsymbol{h}, o, u), \tag{6}$$

**Clipping** distortions refer to the clipped amplitude of audio signals, which are usually caused by low-quality microphones. Clipping can be modeled by restricting signal amplitudes within $[-\eta, +\eta]$:

$$d_{\text{clip}}(\boldsymbol{s}) = \max(\min(\boldsymbol{s}, \eta), -\eta), \eta \in [0, 1]. \tag{7}$$

In the frequency domain, the clipping effect produces harmonic components in the high-frequency part and degrades speech intelligibility accordingly.

## 3 METHODOLOGY

### 3.1 ONE-STAGE SPEECH RESTORATION MODELS

Previous deep learning based speech restoration models are usually in one stage. That is, a model predicts restored speech $\hat{\boldsymbol{s}}$ from input $\boldsymbol{x}$ directly:

$$f : \boldsymbol{x} \rightarrow \hat{\boldsymbol{s}}. \tag{8}$$

The mapping function $f(\cdot)$ can be modeled by time domain speech restoration systems such as one-dimensional convolutional neural networks (Luo & Mesgarani, 2019) or frequency domain systems such as mask-based (Narayanan & Wang, 2013) methods:

$$\hat{\boldsymbol{S}} = (F_{\text{sp}}(|\boldsymbol{X}|; \theta) \odot (|\boldsymbol{X}| + \epsilon))e^{j\angle \boldsymbol{X}}. \tag{9}$$

where $\boldsymbol{X}$ is the short-time fourier transform (STFT) of $\boldsymbol{x}$ and $\epsilon$ is a small positive constant. $\boldsymbol{X}$ has a shape of $T \times F$ where $T$ is the number of frame and $F$ is the number of frequency bins. The output of the mask estimation function $F(\cdot; \theta)$ is multiplied by the magnitude spectrogram $|\boldsymbol{X}|$ to produce the target spectrogram estimation $\hat{\boldsymbol{S}}$. Then, inverse short-time fourier transform (iSTFT) is applied on $\hat{\boldsymbol{S}}$ to obtain $\hat{\boldsymbol{s}}$. The one-stage speech restoration models are typically optimized by minimizing the mean absolute error (MAE) loss between the estimated spectrogram $\hat{\boldsymbol{S}}$ and the target spectrogram $\boldsymbol{S}$:

$$\mathcal{L} = \left\| |\hat{\boldsymbol{S}}| - |\boldsymbol{S}| \right\|_1 \tag{10}$$

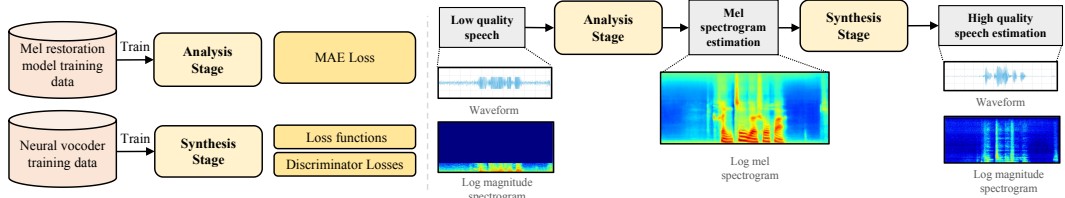

**Figure 2:** Overview of the proposed *VoiceFixer* system.

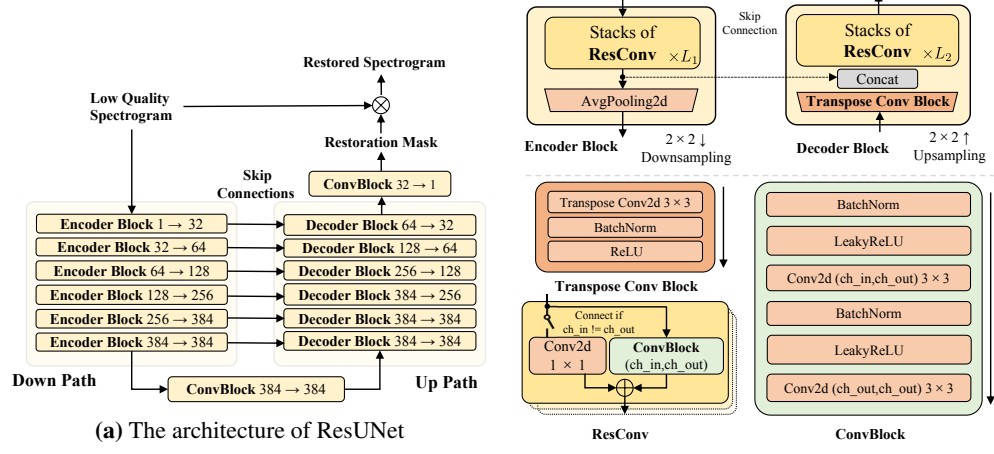

**(a)** The architecture of ResUNet

**(b)** Details of the encoder and decoder blocks of the ResUnet

**Figure 3:** The architecture of ResUNet, which output has the same size as input.

Previous one-stage models usually build on high-dimensional features such as time samples and the STFT spectrograms. However, Kuo & Sloan (2005) point out that the high-dimensional features will lead to exponential growth in search space. The model can work on the high-dimensional features under the premise of enlarging the model capacity but may also fail in challenging tasks. Therefore, it would be beneficial if we could build a system on more delicate low-dimensional features.

## 3.2 VOICEFIXER

In this study, we propose *VoiceFixer*, a two-stage speech restoration framework. Multi-stage methods have achieved state-of-the-art performance in many speech processing tasks (Jarrett et al., 2009; Tan et al., 2020). In speech restoration, our proposed *VoiceFixer* breaks the conventional one-stage system into a two-stage system:

$$f : \boldsymbol{x} \mapsto \boldsymbol{z}, \tag{11}$$

$$g : \boldsymbol{z} \mapsto \hat{\boldsymbol{s}}. \tag{12}$$

Equation 11 denotes the analysis stage of *VoiceFixer* where a distorted speech $\boldsymbol{x}$ is mapped into a representation $\boldsymbol{z}$. Equation 12 denotes the synthesis stage of *VoiceFixer*, which synthesize $\boldsymbol{z}$ to the restored speech $\hat{\boldsymbol{s}}$. Through the two-stage processing, *VoiceFixer* mimics the human perception of speech described in Section 1.

### 3.2.1 ANALYSIS STAGE

The goal of the analysis stage is to predict the intermediate representation $\boldsymbol{z}$, which can be used later to recover the speech signal. In our study, we choose mel spectrogram as the intermediate representation. Mel spectrogram has been widely used in many speech proceessing tasks (Shen et al., 2018; Kong et al., 2019; Narayanan & Wang, 2013). The frequency dimension of mel spectrogram is usually much smaller than that of STFT thus can be regarded as a way of feature dimension

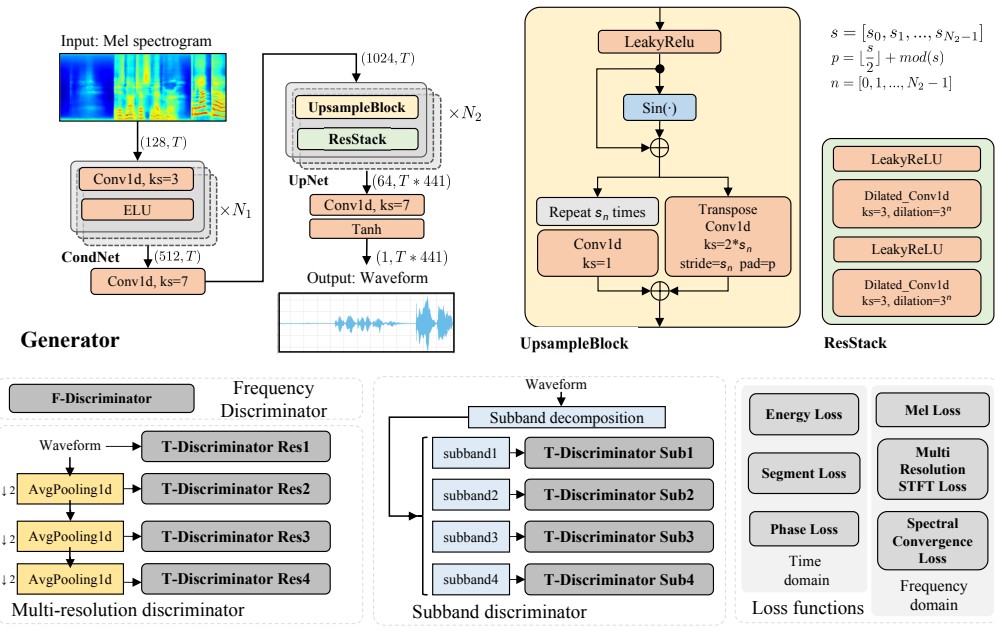

**Figure 4:** The architecture and training scheme of *TFGAN*, whose generator is later used as vocoder. The generator takes mel spectrogram as input and upsampled it into waveform. Both output waveform and its STFT spectrogram are used to compute loss. We employ both time and frequency discriminators for discriminative training.

reduction. So, the objective of the analysis stage becomes to restore mel spectrograms of the target signals. The mel spectrogram restoration process can be written as the following equation,

$$\hat{\boldsymbol{S}}_{\text{mel}} = f_{\text{mel}}(\boldsymbol{X}_{\text{mel}}; \alpha) \odot (\boldsymbol{X}_{\text{mel}} + \epsilon), \tag{13}$$

where $\boldsymbol{X}_{\text{mel}}$ is the mel spectrogram of $\boldsymbol{x}$. It is calculated by $\boldsymbol{X}_{\text{mel}} = |\boldsymbol{X}|\boldsymbol{W}$ where $\boldsymbol{W}$ is a set of mel filter banks with shape of $F \times F'$. The columns of $\boldsymbol{W}$ are not divided by the width of their mel bands, i.e., area normalization, because this will make the restoration model difficult to recover the high-frequency part. The mapping function $f_{\text{mel}}(\cdot; \alpha)$ is the mel restoration mask estimation model parameterized by $\alpha$. The output of $f_{\text{mel}}$ is multiplied by $\boldsymbol{X}_{\text{mel}}$ to predict the target mel spectrogram.

We use ResUNet (Kong et al., 2021a) to model the analysis stage as shown in Figure 3a, which is an improved UNet (Ronneberger et al., 2015). The ResUNet consists of several encoder and decoder blocks. There are skip connections between encoder and decoder blocks at the same level. As is shown in Figure 3b, both encoder and decoder block share the same structure, which is a series of residual convolutions (ResConv). Each convolutional layer in ResConv consists of a batch normalization (BN), a leakyReLU activation, and a linear convolutional operation. The encoder blocks apply average pooling for downsampling. The decoder blocks apply transpose convolution for upsampling. In addition to ResUNet, we implement the analysis stage with fully connected deep neural network (DNN), and bidirectional gated recurrent units (BiGRU) (Chung et al., 2014) for comparison. The DNN consists of six fully connected layers. The BiGRU has similar structures with DNN except for replacing the last two layers of DNN into bi-directional GRU layers.

The details of these three models are discussed in Appendix B.1. We will refer to ResUNet as UNet later for abbreviation. We optimize the model in the analysis stage using the MAE loss between the estimated mel spectrogram $\hat{\boldsymbol{S}}_{\text{mel}}$ and the target mel spectrogram $\boldsymbol{S}_{\text{mel}}$:

$$\mathcal{L}_{\text{ana}} = \left\| \hat{\boldsymbol{S}}_{\text{mel}} - \boldsymbol{S}_{\text{mel}} \right\|_1 \tag{14}$$

### 3.2.2 SYNTHESIS STAGE

The synthesis stage is realized by a neural vocoder that synthesizes the mel spectrogram into waveform as denoted in Equation 15:

$$\hat{s} = g(\boldsymbol{X}_{\text{mel}}; \beta), \tag{15}$$

where $g(\cdot; \beta)$ stands for the vocoder model parameterized by $\beta$. We employ a recently proposed non-autoregressive model, time and frequency domain based generative adversarial network (TFGAN), as the vocoder.

Figure 4 shows the detailed architecture of *TFGAN*, in which the input mel spectrogram $\boldsymbol{X}_{\text{mel}}$ will first pass through a condition network *CondNet*, which contains $N_1$ one-dimensional convolution layers with exponential linear unit activations (Clevert et al., 2015). Then, in *UpNet*, the feature is upsampled $N_2$ times with ratios of $s_0, s_1, ...,$ and $s_{N_2-1}$ using *UpsampleBlock* and *ResStacks*. Within the *UpsampleBlock*, the input is first passed through a leakyReLU activation and then fed into a sinusoidal function, which output is added to its input to remove periodic artifacts in breathing part of speech. Then, the output is bifurcated into two branches for upsampling. One branch repeats the samples $s_n$ times followed by a one-dimensional convolution. The other branch uses a stride $s_n$ transpose convolution. The output of the repeat and transpose convolution branches are added together as the output of *UpsampleBlock*. *ResStacks* module contains two dilated convolution layers with leakyReLU activations. The exponentially growing dilation in *ResStack* enable the model to capture long range dependencies. The *TFGAN* in our synthesis model applies $N_2 = 4$. After four *UpsampleBlock* blocks with ratios $[7, 7, 3, 3]$, each frame of the mel spectrogram is transformed into a sequence with 441 samples corresponding to 10 ms of audio sampled at 44.1 kHz.

The training criteria of the vocoder consist of frequency domain loss $\mathcal{L}_F$, time domain loss $\mathcal{L}_T$, and weighted discriminator loss $\mathcal{L}_D$:

$$\mathcal{L}_{\text{syn}} = \mathcal{L}_F + \mathcal{L}_T + \lambda_D \mathcal{L}_D, \tag{16}$$

The frequency domain loss $\mathcal{L}_F$ is the combination of a mel loss $\mathcal{L}_{\text{mel}}$ and multi-resolution spectrogram losses:

$$\mathcal{L}_F(\hat{s}, s) = \lambda_{\text{mel}} \mathcal{L}_{\text{mel}}(\hat{s}, s) + \sum_{k=1}^{K_F} (\lambda_{\text{sc}} \mathcal{L}_{\text{sc}}^{(k)}(\hat{s}, s) + \lambda_{\text{mag}} \mathcal{L}_{\text{mag}}^{(k)}(\hat{s}, s)) \tag{17}$$

where $\mathcal{L}_{\text{sc}}$ and $\mathcal{L}_{\text{mag}}$ are the spectrogram losses calculated in the linear and log scale, respectively. There are $K_F$ different window sizes ranging from 64 to 4096 to calculate $\mathcal{L}_{\text{sc}}$ and $\mathcal{L}_{\text{mag}}$ so that the trained vocoder is tolerant over phase mismatch (Yamamoto et al., 2020; Juvela et al., 2019). Table 2 in Appendix B.2 shows the detailed configurations.

Time domain loss is complementary to frequency domain loss to address problems such as periodic artifacts. Time domain loss combines segment loss $\mathcal{L}_{\text{seg}}^{(k)}$, energy loss $\mathcal{L}_{\text{energy}}^{(k)}$ and phase loss $\mathcal{L}_{\text{phase}}^{(k)}$:

$$\mathcal{L}_T(\hat{s}, s) = \sum_{k=1}^{K_T} (\lambda_{\text{seg}} \mathcal{L}_{\text{seg}}^{(k)}(\hat{s}, s)) + \lambda_{\text{energy}} \mathcal{L}_{\text{energy}}^{(k)}(\hat{s}, s) + \lambda_{\text{phase}} \mathcal{L}_{\text{phase}}^{(k)}(\hat{s}, s) \tag{18}$$

where segment loss $\mathcal{L}_{\text{seg}}^{(k)}$, energy loss $\mathcal{L}_{\text{energy}}^{(k)}$ and phase loss $\mathcal{L}_{\text{phase}}^{(k)}$ are described in Equation 24, 25, and 26 of Appendix B.2. There are $K_T$ different window sizes ranging from 1 to 960 to calculate time domain loss at different resolutions. The details of window sizes are shown in Table 3 of Appendix B.2. The energy loss and phase loss have the advantage of alleviating metallic sounds.

Discriminative training is an effective way to train neural vocoders (Kong et al., 2020; Kumar et al., 2019). In our study, we utilize a group of discriminators, including a multi-resolution time discriminator $D_T$, a subband discriminator $D_{sub}$, and frequency discriminator $D_F$:

$$D(\boldsymbol{s}) = \sum_{r=1}^{R_T} D_T^{(r)}(\boldsymbol{s}) + D_{sub}(\boldsymbol{s}) + D_F(\boldsymbol{s}) \tag{19}$$

$$\mathcal{L}_D(\boldsymbol{s}, \hat{s}) = \min_g \max_D (\mathbb{E}_{\boldsymbol{s}}(\log(D(\boldsymbol{s}))) + \mathbb{E}_{\hat{s}}(\log(1 - D(\hat{s})))). \tag{20}$$

The multi-resolution discriminators $D_T$ take signals from $R_T$ kinds of time resolutions after average pooling as input. The subband discriminator $D_{sub}$ performs subband decomposition (Liu et al., 2020) on the waveform, producing four subband signals to feed into four *T-discrminators*, respectively. Frequency discriminator $D_F$ takes the linear spectrogram as input and outputs real or fake labels. The bottom part of Figure 4 shows the main idea of *T-discriminator* and *F-discriminator*. Appendix B.2 describes the detailed discriminator architectures.

There are two advantages of using neural vocoder in the synthesis stage. First, neural vocoder trained using a large amount of speech data contains prior knowledge on the structural distribution of speech signals, which is crucial to the restoration of distorted speech. The amount of training data of vocoder is more than that used in conventional SSR methods with limited speaker numbers. Second, the neural vocoder typically takes the mel spectrogram as input, resulting in fewer feature dimensions than the STFT features. The reduction in dimension helps to lower computational costs and achieve better performance in the analysis stage.

## 4 EXPERIMENTS

### 4.1 DATASETS AND EVALUATION METRICS

**Training sets:** The training speech datasets we use including VCTK (Yamagishi et al., 2019), AISHELL-3 (Shi et al., 2020), and HQ-TTS. We call the noise datasets used for training as VD-Noise. To simulate the reverberations, we employ a set of RIRs to create an RIR-44k dataset. We use VCTK, VD-Noise, and the training part of RIR-44k to train the analysis stage. AISHELL-3, VCTK, and HQ-TTS datasets are used to train the vocoder in the synthesis stage. The details of those datasets and the RIRs simulation configurations are discussed in Appendix C.1.

**Test sets:** We employ VCTK-Demand (Valentini-Botinhao et al., 2017) as the denoising test set and name it as DENOISE. We call our speech super-resolution, declipping, and dereverberation evaluation test sets as SR, DECLI, and DEREV, respectively. In addition, we create an ALL-GSR test set containing all distortions. We introduce the details of how we build these test sets in Appendix C.3.

**Evaluation metrics:** The metrics we adopt include log-spectral distance (LSD) (Erell & Weintraub, 1990), wide band perceptual evaluation of speech quality (PESQ-wb) (Rix et al., 2001), structural similarity (SSIM) (Wang et al., 2004), and scale-invariant signal to noise ratio (SiSNR) (Le Roux et al., 2019). We use mean opinion scores (MOS) to subjevtively evaluate different systems.

The output of the vocoder is not strictly aligned on sample level with the target, as is often the case in generative model (Kumar et al., 2020). This effect will degrade the metrics, especially for those calculated on time samples such as the SiSNR. So, to compensate the SiSNR, we design similar metric, scale-invariant spectrogram to noise ratio (SiSPNR), to measure the discrepancies on the spectrograms. Details of the metrics are described in Appendix C.4.

### 4.2 DISTORTIONS SIMULATION

For the SSR task, we perform only one type of distortion for evaluation. For the GSR task, we first assume that $\mathbb{D} = \{d_{\mathrm{noise}}, d_{\mathrm{rev}}, d_{\mathrm{low\_res}}, d_{\mathrm{clip}}\}$ because those distortions are the most common distortions in daily environment (Ribas et al., 2016). Second, we assume that $Q \leq 4$ in Equation 3. In other words, each distortion in $\mathbb{D}$ appears at most one time. Then, we generate the distortions following a specific order $d_{\mathrm{rev}}$, $d_{\mathrm{clip}}$, $d_{\mathrm{low\_res}}$, and $d_{\mathrm{noise}}$. These distortions are added randomly using random configurations.

### 4.3 BASELINE SYSTEMS

Table 5 in Appendix D summarizes all the experiments in this study. We implement several SSR and GSR systems using one-stage restoration models. For the GSR, we train a ResUNet model called *GSR-UNet* with all distortions. For the SSR models, we implement a *Denoise-UNet* for additive noise distortion, a *Dereverb-UNet* for reverberation distortion, a *SR-UNet* for low-resolution distortion, and a *Declip-UNet* for clipping distortion. For the SR task, we also include two state-of-the-art models, *NuWave* (Lee & Han, 2021) and *SEANet* (Li et al., 2021) for comparison. For declipping task, we compare with a state-of-the-art synthesis-based method *SSPADE* (Záviška et al.,

2019a). To explore the impact of model size of the mel restoration model, we setup ResUNets with two sizes, *UNet-S* and *UNet*, which have one and four *ResConv* blocks in each encoder and decoder block, respectively.

## 4.4 EVALUATION RESULTS

**Neural vocoder** To evaluate the performance of the neural vocoder, we compare two baselines. The *Target* system denotes using the perfect $s$ for evaluation. The *Unprocessed* system denotes using distorted speech $x$ for evaluation. The *Oracle-Mel* system denotes using the mel spectrogram of $s$ as input to the vocoder, which marks the performance of the vocoder. As shown in Table 1, the *Oracle-Mel* system achieves a MOS score of 3.74, which is close to the *Target* MOS of 3.95, indicating that the vocoder performs well in the synthesis task.

**Figure 5:** Box plot of the MOS scores on general speech restoration task. Red solid line and green dashed line represent median and mean value.

| Models | PESQ | LSD | SiSPNR | SSIM | MOS |
|--------|------|-----|--------|------|-----|
| Unprocessed | 1.94 | 2.00 | 7.20 | 0.64 | 2.38 |
| Oracle-Mel | 2.52 | 0.91 | 11.73 | 0.74 | 3.74 |
| Target | 4.64 | 0.01 | 110.55 | 1.00 | 3.95 |
| GSR-UNet | **2.67** | 1.01 | **12.19** | **0.79** | 3.37 |
| Denoise-UNet | 2.33 | 1.98 | 9.65 | 0.65 | 2.87 |
| Dereverb-UNet | 1.97 | 1.81 | 8.50 | 0.59 | / |
| VF-DNN | 1.55 | 1.18 | 10.13 | 0.68 | / |
| VF-BiGRU | 1.92 | 1.02 | 10.98 | 0.71 | 3.24 |
| VF-UNet-S | 2.01 | 1.02 | 11.09 | 0.71 | / |
| VF-UNet | 2.05 | **1.01** | 11.14 | 0.71 | **3.62** |

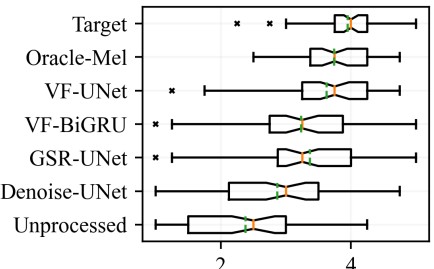

**Table 1:** Average PESQ, LSD, SiSPNR, SSIM and MOS scores on the general speech restoration test set, ALL-GSR, which includes random distortions.

**General speech restorations:** Table 1 shows the evaluation results on ALL-GSR test set. Figure 5 shows the box plot of the MOS scores of these systems. The *GSR-UNet* outperforms the two SSR models, *Denoise-UNet* and *Dereverb-UNet* by a large margin. It surpasses *Denoise-UNet* model by 0.5 on MOS score, which suggests the GSR model is more powerful than the SSR model on this test set. For convenience, we denote *VoiceFixer* as *VF* in tables and figures. We observe that the *VF-UNet* model achieves the highest MOS score and LSD score. Specifically, *VF-UNet* obtains 0.256 higher MOS score than that of *GSR-UNet*. This result indicates that *VoiceFixer* is better than ResUNet based one-stage model on overall quality. What's more, we notice that the MOS score of *VF-UNet* is only 0.11 lower than the *Oracle-Mel*, demonstrating the good performance of the analysis stage. Among the *VoiceFixer* analysis models, the *UNet* front-end achieves the best. The *VF-BiGRU* model achieves similar subjective metrics with the *VF-UNet* model but has much lower MOS scores. This phenomenon shows that the improvement in subjective metrics in *VoiceFixer* is not always consistent with objective evaluation results.

**Super-resolution:** Table 6 in Appendix D.1 shows the evaluation results on the super-resolution test set *SR*. For the 2 kHz, 4 kHz, and 8 kHz to 44.1 kHz super-resolution tasks, *VF-UNet* achieves a significantly higher LSD, SiSPNR and SSIM scores than other models. The LSD value of *VF-UNet* in 2 kHz sampling rate is still higher than the 8 kHz sampling rate score of *GSR-UNet*, *SR-UNet*, *NuWave*, and *SEANet*. This demonstrates the strong performance of *VoiceFixer* on dealing with low sampling rate cases. The *VF-BiGRU* model outperforms *VF-UNet-S* model on average scores for its better performance on low upsample-ratio cases. MOS box plot in Figure 6b shows that *VF-UNet* performs the best on 8 kHz to 44.1 kHz test set. Figure 6e shows the MOS score of *Unprocessed* is close to *Target* on 24 kHz to 44.1 kHz test set, meaning limited perceptual difference between the two sampling rates. On this test set, *SEANet* even achieves a higher MOS score than *Target*. That's due to the high-frequency part it generate contains more energy than that of *Target*, making the results sound clearer.

**Denoising:** We evaluate the speech denoising performance on the DENOISE test set and show results in Table 7 in Appendix D.1. We find that *GSR-UNet* preserves more details in the high-frequency part and has better PESQ and SiSPNR values than the denoising only SSR model *Denoise-UNet*. The reason might be that speech data augmentations and jointly performing super-resolution task can increase the generalization and inpainting ability of the model (Hao et al., 2020). The PESQ

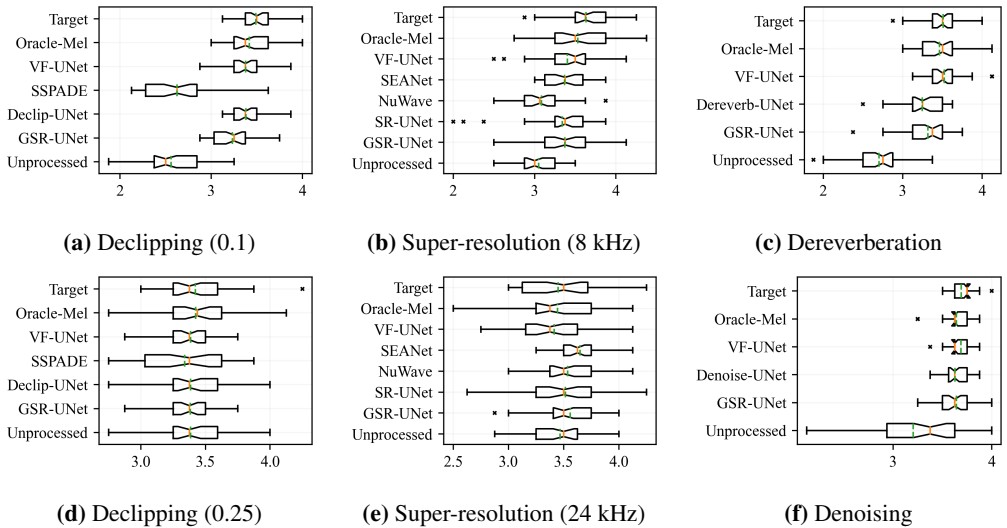

**Figure 6:** Box plot of the MOS scores on speech super-resolution, declipping, dereveberation and denoising.

score of *VF-UNet* reaches 2.43, higher than *SEGAN*, *WaveUNet*, and the model trained with weakly labeled data in Kong et al. (2021b). The MOS evaluations in Figure 6f on speech denoising task also demonstrate that the result of *VF-UNet* sound comparable with one-stage speech denoising models.

**Declipping and dereveberation:** Table 9 and Table 8 in Appendix D.1 show similar performance trends on the speech declipping and speech dereveberation. In both tasks, the SSR model *Dereverb-UNet* and *Declip-UNet* achieve the highest scores. The performance of *GSR-UNet* is slightly worse, but it is acceptable considering that *GSR-UNet* does not need extra training for each task. *SSPADE* performs better on SiSNR, but the PESQ and STOI scores are lower, especially in the 0.1 threshold case. The MOS score in Figure 6d shows that the clipping effect in the 0.25 threshold case is not easy to perceive, leading to high MOS scores across all methods. In Figure 6a, both *Declip-UNet* and *VF-UNet* achieve the highest objective scores on the 0.1 threshold clipping test set. On the dereverberation test set DEREV, *VF-UNet* achieves the highest MOS score 3.52.

## 5 CONCLUSIONS

In this work, we propose *VoiceFixer*, an effective approach for general speech restoration. The evaluation results show that *VoiceFixer* achieves leading performance across the general speech restoration, speech super-resolution, speech denoising, speech dereverberation, and speech declipping tasks. In the future, we will extend *VoiceFixer* to restore general audio signals, including music and general sounds.

## REPRODUCIBILITY STATEMENT

We make our code and datasets anonymously downloadable for painless reproducibility. Our pre-trained *VoiceFixer* and inference code are presented in `https://github.com/anonymous20211004/iclr2022-vf`. The code for performing the experiments discussed in Section 4 is downloadable in `https://github.com/anonymous20211004/iclr2022-train-vf`. The code can be used to conduct evaluations and generate reports on the metrics mentioned in Section 4.1 automatically. The *NuWave* is realized using the code open-sourced by Lee & Han (2021): `https://github.com/mindslab-ai/nuwave`. We reproduce *SSPADE* using the toolbox provided by Záviška et al. (2020) at `https://rajmic.github.io/declipping2020`. Besides, we upload the training and testing sets mentioned in Section 4.1 to `https://zenodo.org/record/5546723`. The AISHELL-3 dataset is open-sourced at `http://www.aishelltech.com/aishell_3`. The HQ-TTS is a collection of datasets from `openslr.org`, as is described in Table 4 of Appendix C.1.

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

# A   APPENDIX A

## A.1   SPEECH RESTORATION TASKS

**Speech super-resolution** A lot of early studies (Nakatoh et al., 2002; Kontio et al., 2007) break super-resolution (SR) into spectral envelop estimation and excitation generation from the low-resolution part. At that time, the direct mapping from the low-resolution part to the high-resolution feature is not widely explored since the dimension of the high-resolution part is relatively high. Later, deep neural network (Li & Lee, 2015; Kuleshov et al., 2017) is introduced to perform SR. These approaches show better subjective quality comparing with traditional methods. To increase the modeling capacity, *TFilm* (Birnbaum et al., 2019) is proposed to model the affine transformation among each time block. Similarly, *WaveNet* also shows effectiveness in extending the bandwidth of a band-limited speech (Gupta et al., 2019). To utilize the information both from the time and frequency domain, Wang & Wang (2021) propose a time-frequency loss that can yield a balanced performance on different metrics. Recently, *NU-GAN* (Kumar et al., 2020) and *NU-Wave* (Lee & Han, 2021) pushed the target sample rate in SR to high fidelity, namely 48 kHz.

Although employing deep neural networks in BWE shows promising results, the generalization capability of these methods is still limited. For example, previous approaches (Kuleshov et al., 2017) usually train and test models with a fixed setup, i.e., fixing the initial and target sampling rates. However, in real-world applications, speech bandwidth is not usually constant. In addition, since the real high-low quality speech pair is hard to collect, typically, we produces low-quality audio with lowpass or bandpass filters during training. In this case, systems tend to overfit specific filters. As discussed in Sulun & Davies (2020), when the kind of filter used during training and testing differ, the performance can fall considerably. To alleviate filter overfitting, Sulun & Davies (2020) propose to train models with multiple kinds of lowpass filters, by which the unseen filters can be handled properly.

**Speech declipping** The methods for speech declipping can be categorized as supervised methods and unsupervised methods. The unsupervised, or blind methods usually perform declipping based on some generic regularization and assumption of what natural audio should look like, such as ASPADE (Kitić et al., 2015), dictionary learning (Rencker et al., 2019), and psychoacoustically motivated l1 minimization (Záviška et al., 2019b). The supervised models, mostly based on deep neural network (DNN) (Bie et al., 2015; Mack & Habets, 2019), are usually trained on clipped and unclipped data pairs. For example, Kashani et al. (2019) treat the declipping as an image-to-image translation problem and utilize the *UNet* to perform spectral mapping. Currently, most of the state-of-the-art methods are unsupervised (Záviška et al., 2020) because they are usually designed to work on different kinds of audio, while the supervised model mainly specialized on data similar to its training data. However, Záviška et al. (2020) believes supervised models still have the potential for better declipping performance.

**Speech denoising** Conventional methods are efficient and effective on dealing with stationary noise, such as spectral substraction (Martin, 1994) and Wiener and Kalman filtering (Kailath, 1981) By comparison, deep learning based models such as *Conv-TasNet* (Luo & Mesgarani, 2019) show higher subjective score and robustness on complex cases. Recently, new schemes have emerged for training speech denoising models. *SEGAN* (Pascual et al., 2017) tried a generative way. Kong et al. (2021b) achieved a denoising model using only weakly labeled data. And Polyak et al. (2021) realized a denoising model using a regeneration approach.

**Speech dereveberation** Some of the early methods in speech dereverberation, such as inverse filtering (Naylor & Gaubitch, 2010) and subband envelope estimation (Wang & Itakura, 1991), aiming at deconvolving the reverberant signal by estimating an inverse filter. However, the inverse filter is hard and not robust to estimate accurately. Other methods, like spectral substraction, is based on an overlap-masking (Nábělek et al., 1989) effect of reverberation. Schwartz et al. (2014) performed dereverberation using Kalman filter and expectation-maximization algorithm. Recently, deep learning based dereverberation methods have emerged as the state-of-the-art. Han et al. (2015) used a fully connected DNN to learn a spectral mapping from reverberant speech to clean speech. In Williamson & Wang (2017), similar to the masking-based denoising methods, they proposed to perform dereverberation using a time-frequency mask.

### A.1.1 Joint restoration and synthetic restoration

**Joint restoration** Many works have adopted the joint restoration approach to improving models. To make the acoustic echo cancellation (AEC) result sound cleaner, *MC-TCN* (Shu et al., 2021) proposed to jointly perform AEC and noise suppression at the same time. *MC-TCN* achieved a mean opinion score of 4.41, outperforming the baseline of AEC Challenge (Cutler et al., 2021) by 0.54. Moreover, in the REVERB challenge (Kinoshita et al., 2013), the test set has both reverberation and noise. Therefore, the proposed methods in the challenge should perform both denoising and dereverberation. In Han et al. (2015), the authors proposed to perform dereverberation and denoising within a single DNN and substantially outperform related methods regarding quality and intelligibility. However, previous joint processing usually involved only two sub tasks. In our study, we joint perform four or more tasks to achieve general restoration.

**Synthetic restoration** Directly estimate the source signal from the observed mixture is hard especially when the SNR is low. Some studies adopted a regeneration approach. In Polyak et al. (2021), the authors utilized an ASR model, a pitch extraction model, and a loudness model to extract semantic level information from the speaker. Then these features were fed to an encode-decoder network to regenerate the speech signal. To maintain the consistency of speaker characteristics, it used an auxiliary identity network to compute the identity feature. Similar to synthetic speech restoration, TTS can be treated as the regeneration of speech from texts.

### A.1.2 Neural vocoder

Vocoder, which maps the encoded speech features to the waveforms, is an indispensable component in speech synthesis. The most widely used input feature for vocoder is mel spectrogram. In recent years, since the emergence of *WaveNet* (Oord et al., 2016), neural network based vocoders demonstrate clear advantages over traditional parametric methods (Morise et al., 2016). Comparing with conventional methods, the quality of *WaveNet* is more closer to the human voice. Later, *WaveRNN* (Yu et al., 2019) is proposed to model the waveform with a single GRU. In this way, *WaveRNN* has much lower complexity comparing with *WaveNet*. However, the autoregressive nature of these models and deep structure make their inference process hard to be paralleled. To address this problem, non-autoregressive models like *WaveGlow* (Prenger et al., 2019) and *WaveFlow* (Ping et al., 2020) were proposed. Afterward, non-autoregressive GAN-based models such as *MelGAN* (Kumar et al., 2019) push the synthesis quality to a comparable level with autoregressive models. Recently, *TFGAN* (Tian et al., 2020) demonstrated strong capability in vocoding. Directed by multiple discriminators and loss functions, *TFGAN* was able to leverage information from both time domain and frequency domain. As a result, the synthesis quality of *TFGAN* is more natural and less metallic comparing with other GAN-based non-autoregressive models. In this work, we realize a universal vocoder based on *TFGAN*, which can reconstruct waveform from mel spectrogram of an arbitrary speaker with good perceptual quality. The pre-trained vocoder is available online to facilitate future studies and reproduce our work.

# B   APPENDIX B

## B.1   DETAILS OF THE ANALYSIS STAGE

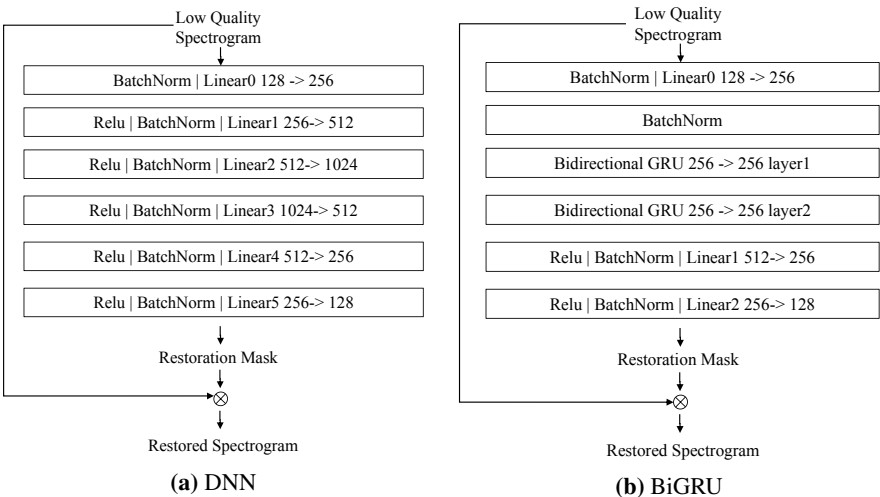

**Figure 7:** The architecture of DNN and Bi-GRU

The DNN and BiGRU we use are shown in Figure 7. DNN is a six layers fully connected network with BatchNorm and ReLU activations. The DNN accept each time step of the low-quality spectrogram as the input feature and output the restoration mask. Similarly, for the BiGRU model, we substitute some layers in DNN to a two-layer bidirectional GRU to capture the time dependency between time steps. To increase the modeling capacity of BiGRU, we expanded the input dimension of GRU to twice the mel frequency dimension with full connected networks.

The detailed architecture of ResUNet is shown in Figure 3a. In the down-path, the input low-quality mel spectrogram will go through 6 encoder blocks, which includes a stack of $L_1$ *ResConv* and a $2 \times 2$ average pooling. In *ResConv*, the outputs of *ConvBlock* and the residual convolution are added together as the output. *ConvBlock* is a typical two layers convolution with BatchNorm and leakyReLU activation functions. The kernel size of residual convolution and the convolution in *ConvBlock* is $1 \times 1$ and $3 \times 3$. Correspondingly, the decoder blocks have the symmetric structure of the encoder blocks. It first performs a transpose convolution with $2 \times 2$ stride and $3 \times 3$ kernels, which result is concatenated with the output of the encoder at the same level to form the input of the decoder. The decoder also contain $L_2$ layers of *ResConv*. The output of the final decoder block is passed to a final *ConvBlock* to fit the output channel.

We use Adam optimizer with $\beta_1 = 0.5, \beta_2 = 0.999$ and a 3e-4 learning rate to optimize the analysis stage of *VoiceFixer*. We treat the first 1000 steps as the warmup phase, during which the learning rate grows linearly from 0 to 3e-4. We decay the learning rate by 0.9 every 400 hours of training data. We perform an evaluation every 200 hours of training data. If we observe three consecutive evaluations with no improvement, we will interrupt the experiment.

For all the STFT and iSTFT, we use the hanning window with a window length of 2048 and a hop length of 441. As all the audio we use is at the 44.1 kHz sample rate, the corresponding spectrogram size in this setting will be $T \times 1025$, where T is the dimension of time frames. For mel spectrogram, the dimensions of the linear spectrogram are transformed into $T \times 128$.

## B.2   DETAILS OF THE SYNTHESIS STAGE

As shown in Table 3, we use 7 kinds of STFT resolutions and 4 kinds of time resolution during the calculation of $\mathcal{L}_F$ and $\mathcal{L}_T$. So $K_F = 7$ in Equation 17 and $K_T = 4$ in Equation 18.

The mel loss $\mathcal{L}_{\mathrm{mel}}$, spectral convergence loss $\mathcal{L}_{\mathrm{sc}}$, STFT magnitude loss $\mathcal{L}_{\mathrm{mag}}$, segment loss $\mathcal{L}_{\mathrm{seg}}$, energy loss $\mathcal{L}_{\mathrm{energy}}$, and phase loss $\mathcal{L}_{\mathrm{phase}}$ are defined in Equation 21 26. The function $v(\cdot)$ is the

| $k$ | 1 | 2 | 3 | 4 | 5 | 6 | 7 |
|---|---|---|---|---|---|---|---|
| win-length | 4096 | 2048 | 1024 | 512 | 256 | 128 | 64 |
| hop-length | 2048 | 1024 | 512 | 256 | 128 | 64 | 32 |
| fft-size | 8192 | 4096 | 2048 | 1024 | 512 | 256 | 128 |

| $k$ | 1 | 2 | 3 | 4 |
|---|---|---|---|---|
| frame-length | 1 | 240 | 480 | 960 |
| hop-length | 1 | 120 | 240 | 480 |

**Table 2:** STFT setup for different $k$ in $\mathcal{L}_F$.

**Table 3:** Windowing setup for different $k$ in $\mathcal{L}_T$.

windowing function that divide time sample into $w$ windows and compute mean value within each window, $v(s)_{1\times w} = (\mathrm{mean}(\boldsymbol{s}_0), \mathrm{mean}(\boldsymbol{s}_1), ..., \mathrm{mean}(\boldsymbol{s}_{w-1}))$. Each $s_w$ stand for windowed $s$. $\Delta$ stand for first difference.

$$\mathcal{L}_{\mathrm{mel}}(\hat{\boldsymbol{s}}, \boldsymbol{s}) = \left\| \hat{\boldsymbol{S}}_{\mathrm{mel}} - \boldsymbol{S}_{\mathrm{mel}} \right\|_2 \tag{21}$$

$$\mathcal{L}_{\mathrm{sc}}(\hat{\boldsymbol{s}}, \boldsymbol{s}) = \frac{\left\| |\hat{\boldsymbol{S}}| - |\boldsymbol{S}| \right\|_F}{\left\| |\hat{S}| \right\|_F} \tag{22}$$

$$\mathcal{L}_{\mathrm{mag}}(\hat{\boldsymbol{s}}, \boldsymbol{s}) = \left\| \log(|\hat{\boldsymbol{S}}|) - \log(|\boldsymbol{S}|) \right\|_1 , \tag{23}$$

$$\mathcal{L}_{\mathrm{seg}}(\hat{\boldsymbol{s}}, \boldsymbol{s}) = \left\| v(\hat{\boldsymbol{s}_w}) - v(\boldsymbol{s}_w) \right\|_1 , \tag{24}$$

$$\mathcal{L}_{\mathrm{energy}}(\hat{\boldsymbol{s}}, \boldsymbol{s}) = \left\| v(\hat{\boldsymbol{s}_w}^2) - v(\boldsymbol{s}_w^2) \right\|_1 , \tag{25}$$

$$\mathcal{L}_{\mathrm{phase}}(\hat{\boldsymbol{s}}, \boldsymbol{s}) = \left\| \Delta v(\hat{\boldsymbol{s}_w}^2) - \Delta v(\boldsymbol{s}_w^2) \right\|_1 , \tag{26}$$

Table 9 and Table 8 show the structure of frequency and time domain discriminators. The subband discriminators $D_{sub}$ and multi-resolution time discriminators $D_T^{(r)}(s)$ use the structure of *T-discriminator*, which is a stack of one dimensional convolution with grouping and large kernal size. The frequency discriminator $D_F$ use the similar module *ResConv* similar to *ResUNet* shown in Figure 3b.

| T-discriminator |
|---|
| Conv1d(1, 128, kernal_size=16), LeakyRelu(0.2) |
| Conv1d(128, 128, kernal_size=41, stride=4, padding=20, groups=8), LeakyRelu(0.2) |
| Conv1d(128, 128, kernal_size=41, stride=4, padding=20, groups=16), LeakyRelu(0.2) |
| Conv1d(128, 128, kernal_size=41, stride=4, padding=20, groups=32), LeakyRelu(0.2) |
| Conv1d(128, 1, kernal_size=3, stride=1, padding=1), LeakyRelu(0.2) |

| F-discriminator |
|---|
| Conv2d(1,32,kernal_size=(3,3)) |
| ResConv(32, 32, stride=1,kernal_size=(3,3)) |
| ResConv(32, 32, stride=1,kernal_size=(3,3)) |
| ResConv(32, 64, stride=2,kernal_size=(3,3)) |
| ResConv(64, 64, stride=1,kernal_size=(3,3)) |
| ResConv(64, 32, stride=2,kernal_size=(3,3)) |
| ResConv(32, 32, stride=1,kernal_size=(3,3)) |
| ResConv(32, 32, stride=2,kernal_size=(3,3)) |
| ResConv(32, 32, stride=1,kernal_size=(3,3)) |

**Figure 8:** The structure of *T-discriminator*.

**Figure 9:** The structure of *F-discriminator*.

For the training of vocoder, we setting up the $\lambda_D$ to $\lambda_{\mathrm{seg}}$ value in Equation 16, Equation 17, and Equation 18 as $\lambda_D = 4.0, \lambda_{\mathrm{mel}} = 50, \lambda_{\mathrm{sc}} = 5.0, \lambda_{\mathrm{mag}} = 5.0, \lambda_{\mathrm{energy}} = 100.0, \lambda_{\mathrm{phase}} = 100.0$, and $\lambda_{\mathrm{seg}} = 200.0$

# C   APPENDIX C

## C.1   DATASETS

**Clean speech** CSTR VCTK corpus (Yamagishi et al., 2019) is a multi-speaker English corpus containing 110 speakers with different accents. We split it into a training part VCTK-Train and a testing part VCTK-Test. The version of VCTK we used is 0.92. To follow the data preparation strategy of Lee & Han (2021), only the *mic1* microphone data is used for experiments, and *p280* and *p315* are omitted for the technical issues. For the remaining 108 speakers, the last 8 speakers, *p360,p361,p362,p363,p364,p374,p376,s5* are splitted as test set VCTK-Test. Within the other 100 speakers, *p232* and *p257* are omitted because they are used later in the test set DENOISE, the remaining 98 speakers are defined as VCTK-Train. Except for the training of *NuWave*, all the utterances are resampled at the 44.1 kHz sample rate. AISHELL-3 is an open-source Hi-Fi mandarin speech corpus, containing 88035 utterances with a total duration of 85 hours. HQ-TTS dataset contains 191 hours of clean speech data collected from a serial of datasets on `openslr.org`. In Table 4, we include the details of *HQ-TTS*, including the URL and language types of each subset.

Table 4: The components of HQ-TTS dataset.

| URL | Languages | URL | Languages |
|---|---|---|---|
| http://www.openslr.org/32/ | Afrikaans, Sesotho, Setswana and isiXhosa | http://www.openslr.org/70/ | Nigerian English |
| http://www.openslr.org/37/ | Bangladesh Bengali and Indian Bengali | http://www.openslr.org/71/ | Chilean Spanish |
| http://www.openslr.org/41/ | Javanese | http://www.openslr.org/72/ | Colombian Spanish |
| http://www.openslr.org/42/ | Khmer | http://www.openslr.org/73/ | Peruvian Spanish |
| http://www.openslr.org/43/ | Nepali | http://www.openslr.org/74/ | Puerto Rico Spanish |
| http://www.openslr.org/44/ | Sundanese | http://www.openslr.org/75/ | Venezuelan Spanish |
| http://www.openslr.org/61/ | Spanish | http://www.openslr.org/76/ | Basque |
| http://www.openslr.org/63/ | Malayalam | http://www.openslr.org/77/ | Galician |
| http://www.openslr.org/64/ | Marathi | http://www.openslr.org/78/ | Gujarati |
| http://www.openslr.org/65/ | Tamil | http://www.openslr.org/79/ | Kannada |
| http://www.openslr.org/66/ | Telugu | http://www.openslr.org/80/ | Gujarati |
| http://www.openslr.org/69/ | Catalan | | |

**Noise data** One of the noise dataset we use come from VCTK-Demand (VD) (Valentini-Botinhao et al., 2017), a widely used corpus for speech denoising and noise-robust TTS training. This dataset contains a training part VD-Train and a testing part VD-Test, in which both contain two noisy set VD-Train-Noisy, VD-Test-Noisy and two clean speech set VD-Train-Clean, VD-Test-Clean. To obtain the noise data from this dataset, we minus each noisy data from VD-Train-Noisy with its corresponding clean part in VD-Train-Clean to get the final training noise dataset VD-Noise. The noise data are all resampled to 44.1 kHz. Another noise dataset we adopt is the TUT urban acoustic scenes 2018 dataset (Mesaros et al., 2018), which is originally used for the acoustic scene classification task of DCASE 2018 Challenge. The dataset contains 89 hours of high-quality recording from 10 acoustic scenes such as airport and shopping mall. The total amount of audio is divided into development DCASE-Dev and evaluation DCASE-Eval parts. Both of them contain audio from all cities and all acoustic scenes.

**Room impulse response** We randomly simulated a collection of Room Impulse Response filters to simulate the 44.1 kHz speech room reverberation using an open-source tool [2]. The meters of height, width, and length of the room are sampled randomly in a uniform distribution $\mathcal{U}(1, 12)$. The placement of the microphone is then randomly selected within the room space. For the placement of the sound source, we first determined the distance to the microphone, which is randomly sampled in a Gaussian distribution $\mathcal{N}(\mu, \sigma^2), \mu = 2, \sigma = 4$. If the sampled value is negative or greater than five meters, we will sample the distance again until it meets the requirement. After sampling the distance between the microphone and sound source, the placement of the sound source is randomly selected within the sphere centered at the microphone. The RT60 value we choose come from the uniform distribution $\mathcal{U}(0.05, 1.0)$. For the pickup pattern of the microphone, we randomly choose from omnidirectional and cardioid types. Finally, we simulated 43239 filters, in which we randomly split out 5000 filters as the test set RIR-Test and named other 38239 filters as RIR-Train.

---

[2] `https://github.com/sunits/rir_simulator_python`

## C.2 Training data

We describe the simulation of training data in Algorithm 1. $\mathbb{S} = \{s^{(0)}, s^{(1)}, ..., s^{(i)}\}$, $\mathbb{N} = \{n^{(0)}, n^{(1)}, ..., n^{(i)}\}$, and $\mathbb{R} = \{r(0), r^{(1)}, ..., r^{(i)}\}$ are the speech dataset, noise dataset, and RIR dataset. We use several helper function to describe this algorithm. randomFilterType($\cdot$) is a function that randomly select a type of filter within butterworth, chebyshev, bessel, and ellipic. Resample($x, o_1, u$) is a resampling function that resample the one dimensional signal $x$ from a original samplerate $o_1$ to the target $u$ samplerate. buildFilter($t, c, o_2$) is a filter design function that return a type $t$ filter with cutoff frequency $c$ and order $o_2$. max($\cdot$),min($\cdot$), and abs($\cdot$) is the element wise maximum, minimum, and absolute value function. mean($\cdot$) calculate the mean value of the input.

We first select a speech utterance $s$, a segment of noise $n$ and a RIR filter $r$ randomly from the dataset. Then with $p_1$ probability, we add the reverberation effect using $r$. And with $p_2$ probability, we add clipping effect with a clipping ratio $\eta$, which is sampled in a uniform distrubution $\mathcal{U}(\eta_{low}, \eta_{high})$. To produce low-resolution effect, after determining the filter type $t$, we randomly sample the cutoff frequency $c$ and order $o$ from the uniform distribution $\mathcal{U}(C_{low}, C_{high})$ and $\mathcal{U}(O_{low}, O_{high})$. Then we perform convolution between $x$ and the type $t$ order $o$ lowpass filter with cutoff frequency $c$. Finally the filtered data will be resampled twice, one is resample to $c * 2$ samplerate and another is resample back to 44.1 kHz. We also perform the same lowpass filtering to the noise signal randomly. This operation is necessary because, if not, the model will overfit the pattern that the bandwidth of noise signal is always different from speech. In this case, the model will fail to remove noise when the bandwidth of noise and speech are similar. For the simulation of noisy environment, we randomly add the noise $n$ into the speech signal $x$ using a random SNR $s \sim \mathcal{U}(S_{low}, S_{high})$. To fit the model with all energy levels, we randomly conduct a $q \sim \mathcal{U}(Q_{low}, Q_{high})$ scaling to the input and target data pair.

In our work, we choose the following parameters to perform this algorithm, $p_1 = 0.25$, $p_2 = 0.25$, $p_3 = 0.5$, $\eta_{low} = 0.06$, $\eta_{high} = 0.9$, $C_{low} = 750$, $C_{high} = 22050$, $O_{low} = 2$, $O_{high} = 10$, $S_{low} = -5$, $S_{high} = 40$, $Q_{low} = 0.3$, $Q_{high} = 1.0$.

---

**Algorithm 1:** Add random distortions to high-quality speech signal $s$

**In:** $s \leftarrow \mathbb{S}; n \leftarrow \mathbb{N}; r \leftarrow \mathbb{R}$
**Out:** The high-quality speech $s$ and its randomly distorted version $x$

    $x = s$;
    with $p_1$ probability:
        $x = x * r$ ;                      /* Convolute with RIR filter */
    with $p_2$ probability:
        $\theta \sim \mathcal{U}(\Theta_{low}, \Theta_{high})$ ;            /* Choose clipping ratio */
        $x = \max(\min(x, \theta), -\theta)$ ;          /* Hard clipping */
    with $p_3$ probability:
        $t = \text{randomFilterType}()$ ;
        $c \sim \mathcal{U}(C_{low}, C_{high})$ ;
        $o \sim \mathcal{U}(O_{low}, O_{high})$ ;        /* Random cutoff and order */
        $x = x * \text{buildFilter}(t, c, o)$ ;        /* Low pass filtering */
        $x = \text{Resample}(\text{Resample}(x, 44100, c * 2), c * 2, 44100)$ ;    /* Resample */
        with $p_4$ probability:
            $n = n * \text{buildFilter}(t, c, o)$ ;  /* Low pass filtering on noise */
            $n = \text{Resample}(\text{Resample}(n, c * 2), 44100)$ ;      /* Resample */
    with $p_5$ probability:
        $s \sim \mathcal{U}(S_{low}, S_{high})$ ;
        $q \sim \mathcal{U}(Q_{low}, Q_{high})$ ;        /* Random SNR and scale */
        $n = \frac{n}{\text{mean}(\text{abs}(n))/\text{mean}(\text{abs}(x))}$ ;  /* Normalize the energy of noise */
        $x = (x + \frac{n}{10^{s/20}})$ ;              /* Add noise */
    $s = qs$ ;                         /* Scaling */
    $x = qx$ ;                         /* Scaling */

---

### C.3 Testing data

Testing data is crucial for the evaluation for each kind of distortion. The testing data we use either come from open-sourced test set or simulated by ourselves.

**Super-resolution** The simulation of the SR test set follows the work of (Kuleshov et al., 2017; Wang & Wang, 2021). The low-resolution and target data pairs are obtained by transforming 44.1 kHz sample rate utterances in target speech data VCTK-Test to a lower sample rate $u$. To achieve that, we first convolve the speech data with an order 8 Chebyshev type I lowpass filter with the $\frac{u}{2}$ cutoff frequency. Then we subsample the signal to $u$ sample rate using polyphase filtering. In this work, to test the performance on different sampling rate settings, $u$ are set at 2 kHz, 4 kHz, 8 kHz, 16 kHz, and 24 kHz. We denote the corresponding five testing set as VCTK-4k, VCTK-4k, VCTK-8k, VCTK-16k, and VCTK-24k, respectively.

**Denoising** For the denoising task, we adopt the open-sourced testing set DENOISE described in Appendix C.1. This test set contains 824 utterances from a female speaker and a male speaker. The type of noise data comprises a domestic noise, an office noise, noise in the transport scene, and two street noises. The test set is simulated at four SNR levels, which are 17.5 dB, 12.5 dB, 7.5 dB, and 2.5 dB. The original data is sampled at 48 kHz. We downsample it to 44.1 kHz to fit our experiments.

**Dereverberation** The test set for dereverberation, DEREV, is simulated using VCTK-Test and RIR-Test. For each utterance in VCTK-Test, we first randomly select an RIR from RIR-Test, then we calculate the convolution between the RIR and utterance to build the reverberant speech. Finally, we build 2937 reverberant and target data pairs.

**Declipping** DECLI, the evaluation set for declipping, is also constructed based on VCTK-Test. We perform clipping on VCTK-Test following the equation in Section 2 and choose 0.25, 0.1 as the two setups for the clipping ratio. This result in two declipping test sets with different levels, each containing 2937 clipped speech and target audios.

**General speech restoration** To evaluate the performance on GSR, we simulate a test set ALL-GSR comprising of speech with all kinds of distortion. The clean speeches and noise data used to build ALL-GSR is VCTK-Test and DCASE-Eval. The simulation procedure of ALL-GSR is almost the same to the training data simulation described in Section 4.2. In total, 501 three seconds long utterances are produced in this test set.

**MOS Evaluation** We select a small portion from the test sets to carry out MOS evaluation for each one. In SR, DECLI, and DEREV, we select 38 utterances out for human ratings. In DENOISE and ALL-GSR, we randomly choose 42 and 51 utterances.

### C.4 Evaluation metrics

**Log-spectral distance** LSD is a commonly used metrics on the evaluation of super-resolution performance (Kumar et al., 2020; Lee & Han, 2021; Wang & Wang, 2021). For target signal $s$ and output estimate $\hat{s}$, LSD can be computed as Equation 27, where $\boldsymbol{S}$ and $\hat{\boldsymbol{S}}$ stand for the magnitude spectrogram of $s$ and $\hat{s}$.

$$\text{LSD}(\boldsymbol{S}, \hat{\boldsymbol{S}}) = \frac{1}{T}\Sigma_{t=1}^{T}\sqrt{\frac{1}{F}\Sigma_{f=1}^{F}\log_{10}(\frac{\boldsymbol{S}(f,t)^2}{\hat{\boldsymbol{S}}(f,t)^2})^2} \tag{27}$$

**Perceptual evaluation of speech quality** PESQ is widely used in speech restoration literature as their evaluation metrics (Pascual et al., 2017; Hu et al., 2020). It was originally developed to model the subjective test commonly used in telecommunication. PESQ provides a score ranging from -0.5 to 4.5 and the higher the score, the better quality a speech has. In our work, we used an open-sourced implementation of PESQ to compute these metrics. Since PESQ only works on a 16 kHz sampling rate, we performed a 16 kHz downsampling to the output 44.1k audio before evaluation.

**Structural similarity** SSIM (Wang et al., 2004) is a metrics in image super-resolution. It addresses the shortcoming of pixel-level metrics by taking the image texture into account. We match the implementation of SSIM in (Wang et al., 2004) with ours and compute SSIM as Equation 28, where $\mu_S$ and $\sigma_S$ is the mean and standard deviation of $S$. $\text{Cov}_{S\hat{S}}$ is the Covariance of $S$ and $\hat{S}$. $\epsilon_1 = 0.01$

and $\epsilon_2 = 0.02$ are two constant used to avoid zero division. Similarity is measured within the $K$ 7*7 blocks divided from $\boldsymbol{S}$ and $\hat{\boldsymbol{S}}$.

$$\text{SSIM}(\boldsymbol{S}, \hat{\boldsymbol{S}}) = \Sigma_{k=1}^{K} \left( \frac{(2\mu_{\boldsymbol{S}_k}\mu_{\hat{\boldsymbol{S}}_k} + \epsilon_1)(2\text{Cov}_{\boldsymbol{S}_k\hat{\boldsymbol{S}}_k} + \epsilon_2)}{(\mu_{\boldsymbol{S}_k}^2 + \mu_{\hat{\boldsymbol{S}}_k}^2 + \epsilon_1)(\sigma_{\boldsymbol{S}_k}^2 + \sigma_{\hat{\boldsymbol{S}}_k}^2 + \epsilon_2)} \right) \tag{28}$$

**Scale-invariant signal to noise ratio** SiSNR (Le Roux et al., 2019) is widely used in speech restoration literatures to compare the energy of a signal to its background noise. SiSNR is calculated between target waveform $\boldsymbol{s}$ and waveform estimation $\hat{\boldsymbol{s}}$:

$$\text{SiSNR}(\boldsymbol{s}, \hat{\boldsymbol{s}}) = 10 * \log_{10} \frac{\|\hat{\boldsymbol{s}}_{target}\|^2}{\|\boldsymbol{e}_{noise}\|^2}, \tag{29}$$

where $\hat{\boldsymbol{s}}_{target} = \frac{<\hat{\boldsymbol{s}}, \boldsymbol{s}>\boldsymbol{s}}{\|\hat{\boldsymbol{s}}\|^2}$. A higher SiSNR indicates less discrepancy between the estimation and target.

**Scale-invariant spectrogram to noise ratio** SiSPNR is a spectral metric similar to SiSNR. They have the similar idea except SiSPNR is computed on the magnitude spectrogram. Given the target spectrogram $\boldsymbol{S}$ and estimation $\hat{\boldsymbol{S}}$, the computation of SiSPNR can be formulated as

$$\text{SiSPNR}(\boldsymbol{S}, \hat{\boldsymbol{S}}) = 10 * \log_{10} \frac{\|\hat{\boldsymbol{S}}_{target}\|^2}{\|\boldsymbol{E}_{noise}\|^2} \tag{30}$$

where $\hat{\boldsymbol{S}}_{target} = \frac{<\hat{\boldsymbol{S}}, \boldsymbol{S}>\boldsymbol{S}}{\|\hat{\boldsymbol{S}}\|^2}$. The scale invariant is guranteed by mean normalization of estimated and target spectrogram.

# D APPENDIX D

**Table 5:** Experiments setup. We list the training and testing sets used for each model's training and evluation. Check mark is used to denote whether a model adopts the framework of *VoiceFixer* and whether it is trained for SSR or GSR task.

| Name | VoiceFixer | SSR | GSR | Training sets | Testing sets |
|------|-----------|-----|-----|---------------|--------------|
| Unprocessed | ✗ | ✗ | ✗ | / | DENOISE; DEREV; SR; DECLI; ALL-GSR; |
| Oracle-Mel | ✗ | ✗ | ✗ | / | DENOISE; DEREV; SR; DECLI; ALL-GSR; |
| Vocoder-TFGAN | ✗ | ✗ | ✗ | VCTK-Train; HQ-TTS; AISHELL-3 | DENOISE; DEREV; SR; DECLI; ALL-GSR; |
| Denoise-UNet | ✗ | ✓ | ✗ | VCTK-Train; VD-Noise; | DENOISE; ALL-GSR; |
| Dereverb-UNet | ✗ | ✓ | ✗ | VCTK-Train; RIR-Train; | DEREV |
| SR-UNet | ✗ | ✓ | ✗ | VCTK-Train; | SR |
| Declip-UNet | ✗ | ✓ | ✗ | VCTK-Train; | DECLI |
| NuWave | ✗ | ✓ | ✗ | VCTK-Train; | SR |
| SEANet | ✗ | ✓ | ✗ | VCTK-Train; | SR |
| SSPADE | ✗ | ✓ | ✗ | / | DECLI |
| GSR-UNet | ✗ | ✗ | ✓ | VCTK-Train; VD-Noise; RIR-Train; | DENOISE; DEREV; SR; DECLI; ALL-GSR; |
| VF-DNN | ✓ | ✗ | ✓ | VCTK-Train; VD-Noise; RIR-Train; | DENOISE; DEREV; SR; DECLI; ALL-GSR; |
| VF-BiGRU | ✓ | ✗ | ✓ | VCTK-Train; VD-Noise; RIR-Train; | DENOISE; DEREV; SR; DECLI; ALL-GSR; |
| VF-UNet-S | ✓ | ✗ | ✓ | VCTK-Train; VD-Noise; RIR-Train; | DENOISE; DEREV; SR; DECLI; ALL-GSR; |
| VF-UNet | ✓ | ✗ | ✓ | VCTK-Train; VD-Noise; RIR-Train; | DENOISE; DEREV; SR; DECLI; ALL-GSR; |

## D.1 EVALUATION RESULTS

**Table 6:** Evaluation results on speech super-resolution test set SR, which includes five sampling rate setup. The metrics is calculated at a target sampling rate of 44.1 kHz

| TRAINING SCHEME | | ONE-STAGE MODELS | | | | VOICEFIXER MODELS | | | | OTHERS | | |
|---|---|---|---|---|---|---|---|---|---|---|---|---|
| SampleRate Up Ratio | Metrics | GSR-UNet | SR-UNet | NuWave | SEANet | VF-DNN | VF-BiGRU | VF-UNet-S | VF-UNet | Unprocessed | Oracle-Mel | Target |
| **2kHz 22.1** | LSD | 1.34 | 1.19 | 1.41 | 1.33 | 1.18 | 1.08 | 1.08 | **1.05** | 3.13 | 0.89 | / |
| | SiSPNR | 11.03 | 10.89 | 9.19 | 9.78 | 10.67 | 11.84 | 11.65 | **12.10** | 9.18 | 13.65 | / |
| | SSIM | 0.75 | 0.77 | 0.73 | 0.72 | 0.75 | 0.77 | 0.78 | **0.78** | 0.68 | 0.85 | / |
| **4kHz 11.0** | LSD | 1.27 | 1.18 | 1.35 | 1.24 | 1.15 | 1.03 | 1.04 | **1.02** | 2.97 | 0.89 | / |
| | SiSPNR | 11.48 | 11.10 | 9.65 | 10.58 | 11.07 | 12.27 | 11.98 | **12.41** | 9.52 | 13.65 | / |
| | SSIM | 0.77 | 0.78 | 0.76 | 0.72 | 0.75 | 0.79 | 0.79 | **0.79** | 0.71 | 0.85 | / |
| **8kHz 5.5** | LSD | 1.21 | 1.11 | 1.24 | 1.20 | 1.06 | 0.99 | 1.01 | **0.99** | 2.70 | 0.89 | / |
| | SiSPNR | 12.07 | 11.82 | 10.73 | 11.11 | 11.94 | 12.68 | 12.34 | **12.74** | 9.93 | 13.65 | / |
| | SSIM | 0.81 | **0.82** | 0.80 | 0.74 | 0.78 | 0.81 | 0.81 | 0.81 | 0.76 | 0.85 | / |
| | MOS | 3.37 | 3.34 | 3.09 | 3.37 | / | / | / | **3.40** | 3.05 | 3.53 | 3.63 |
| **16kHz 2.8** | LSD | 1.10 | 0.99 | 1.18 | 1.16 | 1.01 | **0.94** | 0.96 | 0.94 | 2.32 | 0.89 | / |
| | SiSPNR | 13.02 | 13.01 | 11.54 | 11.90 | 12.37 | 13.14 | 12.70 | **13.14** | 10.08 | 13.65 | / |
| | SSIM | 0.85 | **0.88** | 0.81 | 0.75 | 0.82 | 0.82 | 0.82 | 0.82 | 0.83 | 0.85 | / |
| **24kHz 1.8** | LSD | 0.97 | **0.91** | 1.12 | 1.15 | 0.93 | 0.91 | 0.94 | 0.92 | 1.91 | 0.89 | / |
| | SiSPNR | **13.96** | 13.81 | 11.63 | 12.58 | 13.21 | 13.38 | 12.86 | 13.38 | 10.40 | 13.65 | / |
| | SSIM | 0.87 | **0.91** | 0.81 | 0.75 | 0.84 | 0.83 | 0.83 | 0.84 | 0.89 | 0.85 | / |
| | MOS | 3.56 | 3.52 | 3.54 | **3.65** | / | / | / | 3.41 | 3.47 | 3.44 | 3.45 |
| **Average Score** | LSD | 1.18 | 1.07 | 1.26 | 1.21 | 1.07 | 0.99 | 1.01 | **0.98** | 2.61 | 0.89 | / |
| | SiSPNR | 12.31 | 12.13 | 10.55 | 11.19 | 11.85 | 12.66 | 12.31 | **12.75** | 9.82 | 13.65 | / |
| | SSIM | 0.81 | **0.83** | 0.79 | 0.74 | 0.79 | 0.80 | 0.81 | 0.81 | 0.77 | 0.85 | / |

**Table 7:** Evaluation result on the speech denoising test set DENOISE

| Models | SiSNR | PESQ | SiSPNR | MOS |
|--------|-------|------|--------|-----|
| Unprocessed | 8.40 | 1.97 | 9.78 | 3.20 |
| Oracle-Mel | -17.52 | 2.85 | 12.84 | 3.64 |
| Target | / | / | / | 3.69 |
| SEGAN (Pascual et al., 2017) | / | 2.16 | / | / |
| Wave-U-Net (Macartney & Weyde, 2018) | / | 2.40 | / | / |
| Weakly Labelled (Kong et al., 2021) | / | 2.28 | / | / |
| GSR-UNet | 16.42 | **2.82** | **12.25** | 3.64 |
| Denoise-UNet | **17.58** | 2.71 | 11.82 | 3.63 |
| VF-DNN | / | 1.71 | 10.93 | / |
| VF-BiGRU | / | 2.29 | 11.72 | / |
| VF-UNet-S | / | 2.33 | 11.19 | / |
| VF-UNet | / | 2.43 | 11.71 | **3.69** |

**Table 8:** Evaluation results on the speech dereverberation test set DEREV

| Models | PESQ | SiSPNR | MOS |
|--------|------|--------|-----|
| Unprocessed | 1.99 | 14.58 | 2.70 |
| Oracle-Mel | 2.36 | 13.65 | 3.46 |
| Target | / | / | 3.51 |
| GSR-UNet | 2.35 | 14.10 | 3.32 |
| Dereverb-UNet | **2.49** | **14.99** | 3.25 |
| VF-DNN | 1.41 | 11.70 | / |
| VF-BiGRU | 1.69 | 13.00 | / |
| VF-UNet-S | 1.78 | 12.80 | / |
| VF-UNet | 1.86 | 13.21 | **3.52** |

## D.2 ANALYSIS STAGE PERFORMANCE

In this section, we report the mel spectrogram restoration score on different test sets. They are used to evaluate the performane of the analysis stage. We calculate the LSD, SiSPNR, and SSIM values.

**Table 9:** Evaluation results on the speech declipping test set DECLI

| Clipping Level | 0.25 | | | | 0.10 | | | | Average | | | |
|---|---|---|---|---|---|---|---|---|---|---|---|---|
| Models | SiSNR | STOI | PESQ | MOS | SiSNR | STOI | PESQ | MOS | SiSNR | STOI | PESQ | MOS |
| Unprocessed | 9.60 | 0.95 | 2.38 | 2.56 | 4.00 | 0.89 | 1.51 | 2.72 | 6.80 | 0.92 | 1.95 | 2.64 |
| Oracle-Mel | -19.94 | 0.81 | 2.36 | 3.44 | -19.94 | 0.81 | 2.36 | 3.42 | -19.94 | 0.81 | 2.36 | 3.43 |
| Target | / | / | / | 3.42 | / | / | / | 3.49 | / | / | / | 3.46 |
| GSR-UNet | 11.01 | 0.97 | 3.54 | 3.38 | 7.47 | 0.94 | 2.89 | 3.23 | 9.24 | 0.95 | 3.21 | 3.31 |
| Declip-UNet | 12.45 | **0.99** | **3.98** | 3.38 | 8.43 | **0.96** | **3.40** | 3.38 | 10.44 | **0.98** | 3.69 | **3.38** |
| SSPADE | **17.43** | 0.98 | 3.55 | 3.34 | **10.31** | 0.92 | 2.12 | 2.63 | **13.87** | 0.95 | 2.84 | 2.98 |
| VF-DNN | / | 0.76 | 1.72 | / | / | 0.72 | 1.48 | / | / | 0.74 | 1.60 | / |
| VF-BiGRU | / | 0.81 | 2.09 | / | / | 0.79 | 1.82 | / | / | 0.80 | 1.95 | / |
| VF-UNet-S | / | 0.82 | 2.13 | / | / | 0.80 | 1.85 | / | / | 0.81 | 1.99 | / |
| VF-UNet | / | 0.82 | 2.21 | 3.38 | / | 0.80 | 1.93 | 3.38 | / | 0.81 | 2.07 | **3.38** |

The *Unprocessed* column is calculated using the target and unprocessed mel spectrogram. And the *Oracle-Mel* column is calculated using the target spectrogram and itself.

**Table 10:** The Performance of Mel Spectrogram Restroation on DENOISE, DEREV, and ALL-GSR test sets

| | DENOISE | | | DEREV | | | ALL-GSR | | |
|---|---|---|---|---|---|---|---|---|---|
| Models | LSD | SiSPNR | SSIM | LSD | SiSPNR | SSIM | LSD | SiSPNR | SSIM |
| Unprocessed | 1.31 | -1.41 | 0.57 | 0.84 | 10.02 | 0.63 | 1.65 | -3.90 | 0.47 |
| VF-DNN | 0.76 | 7.61 | 0.69 | 0.93 | 8.86 | 0.59 | 0.87 | 6.26 | 0.58 |
| VF-BiGRU | 0.55 | 10.98 | 0.79 | 0.56 | 12.91 | 0.75 | 0.59 | 10.49 | 0.70 |
| VF-UNet-S | 0.52 | 10.29 | 0.82 | 0.47 | 13.61 | **0.82** | 0.55 | 11.08 | 0.75 |
| VF-UNet | **0.46** | **12.27** | **0.84** | **0.46** | **14.89** | 0.82 | **0.53** | **11.36** | **0.76** |

Table 10 shows that on DENOISE, DEREV, and ALL-GSR, all four *VoiceFixer* based models are effective on the restoration of mel spectrogram. Among the four analysis stage models, *UNet* is consistently better than the other three models.

**Table 11:** The performance of mel spectrogram restroation on the SR test set

| SampleRate Upsampling Ratio | Metrics | MODELS | | | | | |
|---|---|---|---|---|---|---|---|
| | | VF-DNN | VF-BiGRU | VF-UNet-S | VF-UNet | Unprocessed | Oracle-Mel |
| 2kHz 22.1 | LSD | 0.80 | 0.68 | 0.65 | **0.60** | 2.99 | 0.00 |
| | SiSPNR | 8.02 | 9.62 | 9.82 | **11.32** | 2.54 | 127.43 |
| | SSIM | 0.56 | 0.63 | 0.66 | **0.68** | 0.40 | 1.00 |
| 4kHz 11.0 | LSD | 0.68 | 0.54 | 0.55 | **0.50** | 2.54 | 0.00 |
| | SiSPNR | 9.66 | 12.23 | 11.22 | **12.83** | 3.16 | 127.43 |
| | SSIM | 0.65 | 0.72 | 0.74 | **0.76** | 0.51 | 1.00 |
| 8kHz 5.5 | LSD | 0.51 | **0.40** | 0.46 | 0.42 | 2.02 | 0.00 |
| | SiSPNR | 12.53 | **14.85** | 12.67 | 14.20 | 4.26 | 127.43 |
| | SSIM | 0.77 | 0.82 | 0.83 | **0.84** | 0.64 | 1.00 |
| 16kHz 2.8 | LSD | 0.43 | **0.26** | 0.37 | 0.33 | 1.53 | 0.00 |
| | SiSPNR | 13.62 | **19.00** | 14.07 | 16.13 | 5.64 | 127.43 |
| | SSIM | 0.83 | 0.91 | 0.90 | **0.91** | 0.77 | 1.00 |
| 24kHz 1.8 | LSD | 0.29 | **0.18** | 0.31 | 0.27 | 1.16 | 0.00 |
| | SiSPNR | 17.94 | **22.16** | 15.53 | 18.59 | 7.40 | 127.43 |
| | SSIM | 0.92 | 0.95 | 0.94 | **0.95** | 0.86 | 1.00 |
| Average | LSD | 0.54 | **0.41** | 0.47 | 0.43 | 2.05 | 0.00 |
| | SiSPNR | 12.35 | **15.57** | 12.66 | 14.61 | 4.60 | 127.43 |
| | SSIM | 0.75 | 0.80 | 0.81 | **0.83** | 0.64 | 1.00 |

Table 11 lists the mel restoration performance on different sampling rates. Although *VF-BiGRU* has fewer parameters than *VF-UNet*, it still achieved the highest score on LSD and SiSPNR averagely. This result indicates that the recurrent structure might be more suitable for the mel spectrogram super-resolution task when the initial sampling rate is high.

## D.3 DEMOS

In this section, we provide some restoration demos using our proposed *VoiceFixer*. In Figure 12, we provide eight restoration demos using our *VF-UNet* model. All the audios used in our demo are either collected from the internet or recorded by ourselves. In each example, the left is the unprocessed spectrogram, and the right is the restored one. After restoration, these seriously distorted speech signals can be revert to relatively high quality.

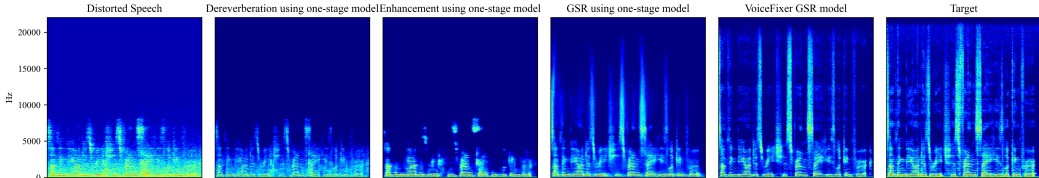

**Figure 10:** Comparison between different restoration mothods. The unprocessed speech is noisy, reverberant, and in low-resolution. The leftmost spectrogram is the unprocessed low-quality speech and the rightmost is the target high-quality spectrogram. In the middle, from left to right, the figures show results processed by one-stage SSR dereveberation model, SSR denoising model, GSR model and *VoiceFixer* based GSR model.

Figure 12b is the speech we recorded using Adobe Audition. We set the sampling rate to 8 kHz and manually add the clipping effect. It also contains some low-frequency noise and reverberation introduced by the recording device and environment. Figure 12a is a speech delivered by *Amelia Earhart*[3], 1897-1937, appeared in the Library of Congress, United States. The original version sounds like a mumble. Figure 12f comes from an interview in a TV news program, which includes multiple distortions. Figure 12e is collected from the audio uploaded by a Youtuber[4]. Probably due to the recording device, her speech is deteriorated seriously by noise, and the energy of speech in the low-frequency part is also relatively low. Figure 12c is the restoration of a Chinese famous old movie *railroad guerrilla*[5], which speech has limited bandwidth. The audio in Figure 12d is selected from a well-known TV series in China, *romance of the three kindoms*[6], in which some parts of the spectrogram are masked off due to the previous audio compression. Figure 12g is a recording[7] selected from a speech delivered by *Sun Yat-sen*, 1866-1925, which is in extremely low-resolution and includes multiple unknown distortions. Figure 12h shows the result of a subway broadcasting we recorded in Shanghai. The low-frequency part of speech almost lost completely, and the reverberation is serious.

To sum up, all these examples showcase the effectiveness of the *VoiceFixer* model on GSR. And to our surprise, it can generate will on unseen distortions such as the spectrogram lost in Figure 12c, Figure 12f, and Figure 12d. In addition, Figure 12e shows that *VoiceFixer* is effective for the compensation of low-frequency energy, making speech sound less machinery and distant. Last but not least, despite the abnormal harmonic structure in the low-frequency part in Figure 12g, our proposed model can still restore its sound, which display the advantages of utilizing the prior knowledge of vocoder.

---

[3] https://www.loc.gov/item/afccal000004
[4] https://v.ixigua.com/egVW74E/
[5] https://www.youtube.com/watch?v=R8lY1qn2CHA
[6] https://www.youtube.com/watch?v=6h7N4Cl1lTw
[7] https://www.bilibili.com/video/BV1WW411V7fR

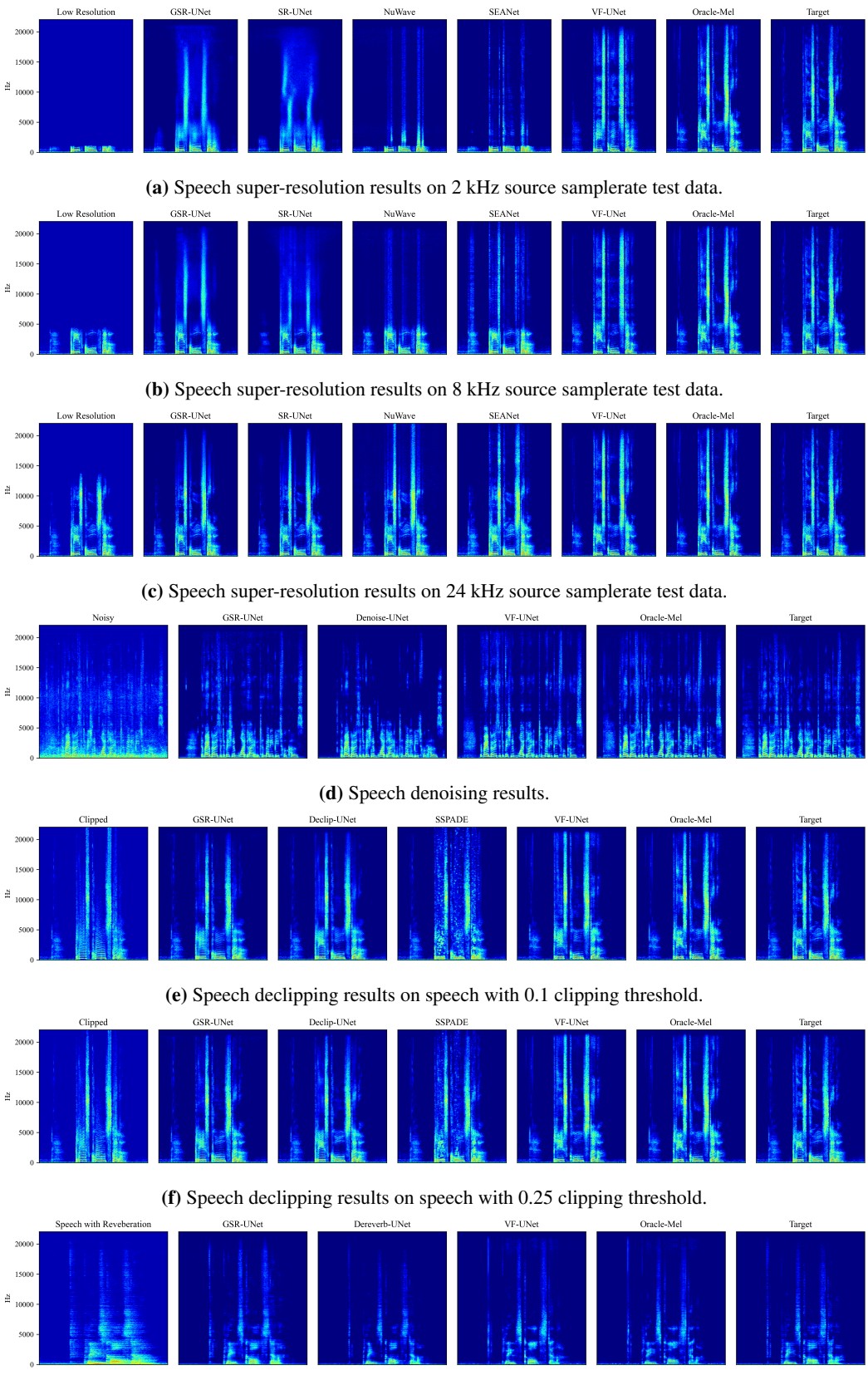

(a) Speech super-resolution results on 2 kHz source samplerate test data.

(b) Speech super-resolution results on 8 kHz source samplerate test data.

(c) Speech super-resolution results on 24 kHz source samplerate test data.

(d) Speech denoising results.

(e) Speech declipping results on speech with 0.1 clipping threshold.

(f) Speech declipping results on speech with 0.25 clipping threshold.

(g) Speech dereverberation results.

**Figure 11:** Comparison between different model on four different tasks using simulated data.

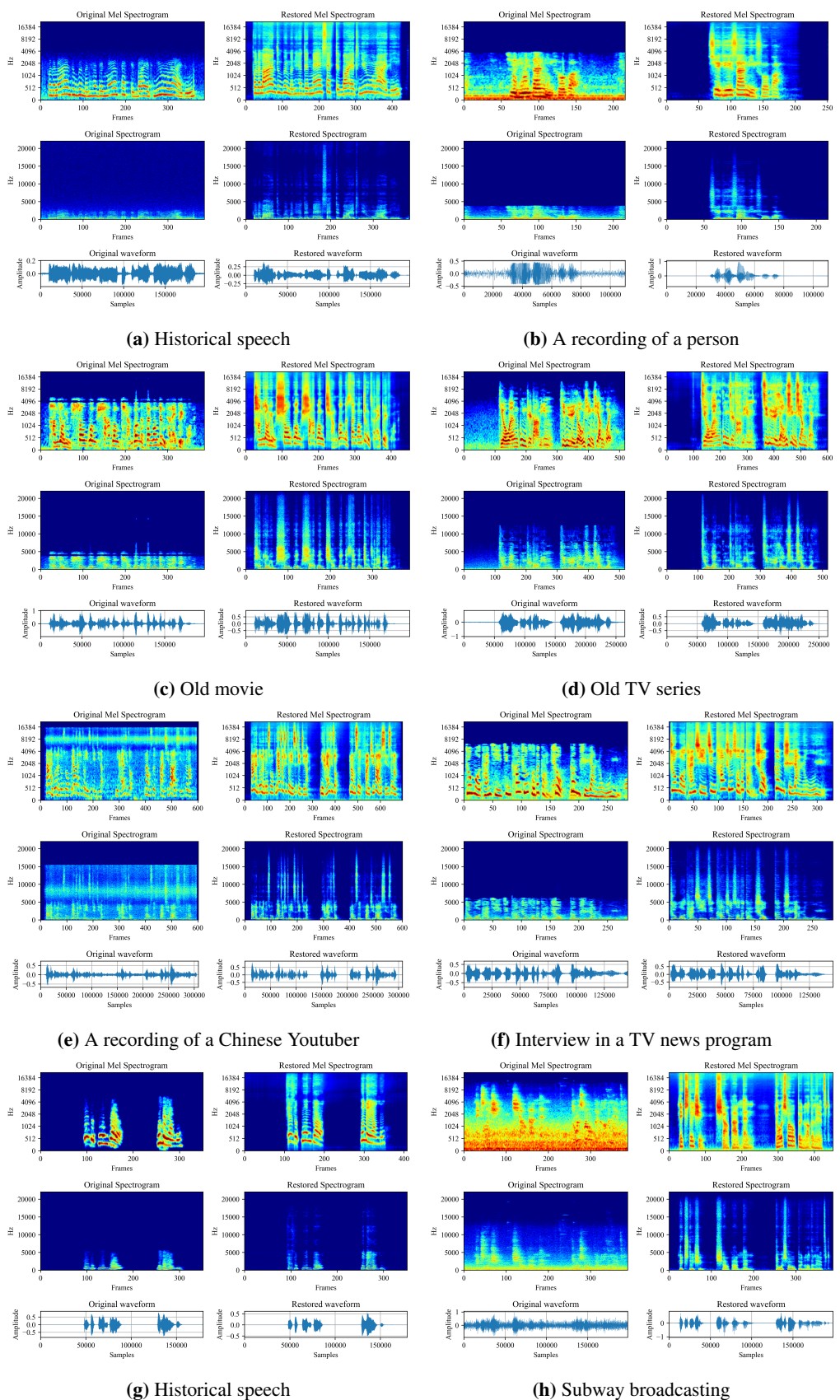

**Figure 12:** Restoration on the audios either collected from the internet or recorded by ourselves.

