# OpenReview forum: "VoiceFixer: Toward General Speech Restoration with Neural Vocoder"
_ICLR.cc/2022/Conference — ICLR 2022 Submitted_

### Official Review · Reviewer_AG4n · 2021-10-27

**Correctness:** 2
**Technical Novelty And Significance:** 1
**Empirical Novelty And Significance:** 2
**Recommendation:** 3
**Confidence:** 5

**Main Review:**

Strengths & weaknesses:
- [$+$] Considering the more general speech enhancement task rather than isolated tasks is an existing effort that deserves more attention.
- [$-$] Considering more general speech enhancement tasks is not novel, as claimed in the paper.
- [$+$] Great effort in the literature review for single-task models.
- [$-$] Misses a substantial amount of prior work for the multiple-task models (see below). Also in result evaluation of, for instance, the denoising task.
- [$-$] Reusing existing architectures and training procedures. Lack of novelty from a machine learning perspective.
- [$-$] No assessment of the effectiveness of the system components, specially the different losses used at different stages.
- [$+$] Great effort in considering separate evaluations for the different tasks.
- [$-$] Results fail at showing improvement in some tasks. Some baselines for the denoising task are not appropriately chosen.
- [$-$] No details on subjective evaluation.
- [$+$] Great effort on reproducibility, both on the data and code sides.
- [$-$] Positioning of the paper should be changed.
- [$-$] Writing and presentation should be improved.

Detailed major issues:
1. The paper positions itself as "propose[ing] a general speech restoration (GSR) task" (example sentence from the Abstract). However, the task has been proposed for quite some time now; the only difference is that perhaps the 4 specific distortions considered in the paper have not been considered together by previous work. Some examples of previous systems performing more than one "speech restoration" task are [1], considering clipping, gaps, codecs, [2], considering BWE, clipping, gaps and whispering, [3], considering denoising, dereverb and equalization, and [4-9], considering denoising and dereverberation. To my knowledge, one of the earliest references to the general task of considering multiple speech distortions dates back to 2019 [2].
1. In my opinion, the paper contains many unsupported claims. One example is in Sec 1: "To address the mismatch problem... restoration (GSR)" (mismatch problem is not characterized, formally defined or empirically proven; GSR is not shown to solve the mismatch problem, actually no mismatch data is considered). Another example is in Sec 3.2: "Through the two-stage processing... human perception of speech" (no empirical result; unclear motivation). Another example is in Sec 3.2.2: "The energy loss and phase loss... metallic sounds" (this is not demonstrated by any result or ablation; also authors do not provide any intuitive motivation). Another example is in Sec 4.1: "we design a similar metrics... discrepancies on the spectrograms" (it not shown anywhere that this new metric really helps in comparing generative vs non-generative models, actually results in Table 7 show this is not the case).
1. In a number of places, the paper tries to establish a relation between a two-stage strategy of analysis/synthesis and "biological mechanisms". I personally think this is far-fetched, misleading and unnecessary. Biological mechanisms are poorly introduced, it is hard to see how model design derives from such biological mechanisms, no experiment is performed to validate such relation, etc.
1. Using a two-stage model, which to my understanding is presented as part of the novelty of the paper, is a common strategy among speech enhancement systems that tackle more than one distortion (see refs. [4-9]). Usually, those are already composed of an analysis plus a synthesis step using a vocoder. Even other papers on denoising employ these two stages (see for example [10]).
1. Architectures for the analysis and synthesis modules are known and the authors do not perform any substantial modification (or at least they do not mention them nor evaluate their worth).
1. Most losses used for training are quite known and some of them are standard. The system uses a large combination of those losses and, besides not providing specific motivation in most of the cases, no results are reported assessing the need of such losses or the impact that they have in final performance. It thus results into a complicated training procedure which may be even unnecessary, and provides no learnings for the speech community.
1. No details are given regarding subjective tests (note that authors sometimes confuse objective with subjective tests, see below). How many listeners? Were they experts? Were they native speakers? Which procedure? Which stimuli? How many of them? Was there some randomization? Specifying the procedure followed in subjective testing is an important point that, if wrong or not specified, by itself alone justifies rejection for a paper in a prestigious venue like ICLR.
1. Distortions are considered to follow a specific order. However, I have serious doubts of whether this is a realistic consideration. Moreover, if the system really aims at generalization, shouldn't we consider all possible orderings even if they are artificial? (thinking here of human generated signals like podcasts for instance).
1. I don't think it makes much sense to compare SSR models with GSR ones in the GSR task (as done in Table 1 for instance). It is good that authors compare SSR/GSR models in the SSR tasks, but doing it on GSR is unfair.
1. In my opinion, most of the results in SSR tasks do not show a significant improvement in terms of MOS. Plots (a,d,e,f) in Fig 6 show that existing systems or simpler counterparts of the proposed system achieve comparable or even better MOS. Only in the tasks of super-resolution and dereverberation the proposed system seems to have a winning edge. This is not bad per se, but puts into perspective the value of the approach.
1. Competitors for the task of denoising are poorly chosen. There are many systems performing much better than the ones reported as baselines in Table 7, both on the supervised and generative side. Some examples are [11-15].
1. Some parts of the writing contain typos or could be better phrased.

Minor comments:
- Abstract: "However, SSR systems..." --> Almost same sentence as before.
- Page 1, paragraph 1: "due to speech" --> since
- I found Sec 2 unnecessary. It could have been moved to the appendix (and this would have left some space to move more interesting parts of the appendix, like some results to the main paper).
- The used vocoder seems to be deterministic (not generative), as it does not have a noise source. It is just an adversarial network, but not generative.
- I personally find that the AISHELL data set contains a lot of reverb and is of rather low quality. I wonder why can this be useful to train the model to do dereverberation when AISHELL is used as clean data.
- Sec 4.4, GSR paragraph: There seems to be a confusion between objective metrics (PESQ, LSD, etc) and subjective metrics (MOS). The authors swap the terms objective and subjective.
- Some references are incomplete (missing page numbers, journal/conference name, book publisher, etc).

References:
1. https://ieeexplore.ieee.org/document/9414721
1. http://dx.doi.org/10.21437/Interspeech.2019-2688
1. https://pixl.cs.princeton.edu/pubs/Su_2021_HSS/Su-HiFi-GAN-2-WASPAA-2021.pdf
1. https://arxiv.org/pdf/2006.00687v1.pdf
1. http://arxiv.org/abs/2102.00429
1. https://arxiv.org/abs/2011.05038
1. https://arxiv.org/abs/2011.03955
1. https://gfx.cs.princeton.edu/pubs/Su_2019_PM/Su_2019_enhancement.pdf
1. https://arxiv.org/abs/2006.05694
1. https://arxiv.org/pdf/2004.04001.pdf
1. https://arxiv.org/pdf/2010.11860v1.pdf
1. https://arxiv.org/abs/2104.03538
1. https://www.isca-speech.org/archive/pdfs/interspeech_2021/hsieh21_interspeech.pdf
1. https://arxiv.org/pdf/2006.12847.pdf
1. https://www.isca-speech.org/archive/pdfs/interspeech_2021/kim21h_interspeech.pdf

**Summary Of The Paper:**

The paper proposes a single system to deal with the speech enhancement tasks of denoising, dereverb, bandwidth extension (BWE) and declipping. The system is a two-stage system composed of an analysis module producing mel-band masks and a synthesis module using a vocoder. Both modules reuse existing architectures. Results comparing the proposed system to author-derived counterparts of it and to some existing systems for specific tasks show improvement for the proposed system in some cases, while achieving similar performance as existing systems in other cases. Data and code for reproducibility are provided.

**Summary Of The Review:**

I think that the paper is not at the level of ICLR standards. Although it could be considered that it achieves interesting results in an existing but rather unexplored task, it contains almost no learnings for the ICLR audience as there seem to be no novel machine learning parts. In this sense, a more specialized venue could be perhaps more appropriate (e.g., InterSpeech, ICASSP, WASPAA). However, in my opinion, both results and writing/presentation also have serious problems that should be solved before resubmitting to such venues.

---

> ### Author Response · Authors · 2021-11-20
> **Thanks for your detailed comments!**
>
> Thanks for your valuable comments. I've modified the minor typos. And I'd like to make some clarifications.
>
> > 9. I don't think it makes much sense to compare SSR models with GSR ones in the GSR task (as done in Table 1 for instance). It is good that authors compare SSR/GSR models in the SSR tasks, but doing it on GSR is unfair.
>
> Yes, comparing these two kinds of models on the GSR test set is unfair to SSR models. The underlying motivation for us is that the real-world data usually contain multiple distortions, just like the GSR test set. So by comparing GSR and SSR on the GSR test set, we'd like to demonstrate GSR task is more practical than the SSR task.
>
> > 10. In my opinion, most of the results in SSR tasks do not show a significant improvement in terms of MOS. Plots (a,d,e,f) in Fig 6 show that existing systems or simpler counterparts of the proposed system achieve comparable or even better MOS. Only in the tasks of super-resolution and dereverberation the proposed system seems to have a winning edge. This is not bad per se, but puts into perspective the value of the approach.
>
> In our opinion, being able to build a GSR model that can have on par performance with each SSR model is already a valuable output in our research. Because the GSR model needs to handle multiple tasks simultaneously, which would surely lower its performance on each task compared with trained in the SSR way. Meanwhile, on the GSR test set, we also demonstrated our approach has better performance than the conventional ResNet model.
>
> > I personally find that the AISHELL data set contains a lot of reverb and is of rather low quality. I wonder why can this be useful to train the model to do dereverberation when AISHELL is used as clean data.
>
> AISHELL data is only used for the training of vocoder, which does not involve dereverberation.

---

> > ### Comment · Reviewer_AG4n · 2021-11-26
> > **Thanks**
> >
> > Thanks for your answer. After considering that and the other reviews/answers I will keep my score.

---

### Official Review · Reviewer_Q5gX · 2021-11-01

**Correctness:** 2
**Technical Novelty And Significance:** 2
**Empirical Novelty And Significance:** 2
**Recommendation:** 5
**Confidence:** 4

**Main Review:**

First of all, I would like to mention that this is a well written paper which discusses an interesting idea for addressing the problem of speech distortions in a more general way. The paper also details many experimental design choices that researchers would need to replicate the results, which is a strong plus. It is also important that the authors have conducted subjective evaluation experiments to show how their algorithm performs wrt the opinion of various humans. Overall, I believe that the paper has great potential if some concerns are addressed. Please see below my concerns and questions to the authors in descending order of importance (0 is the most important):

0. The main problem of this paper stems from the limited originality of the idea behind the general speech restoration method proposed. Specifically, the generative setup where a generative network tries to capture the distribution of clean speech and generate relevant frames has been recently proposed in [1] and shown great potential for speech enhancement. On the other hand, breaking the optimization process in two distinct steps for operating on a feature space where the speech restoration could be easier to obtain has been proposed in [2] for speech and universal audio separation tasks. Although the combination of the two ideas is interesting, the paper does not sufficiently discuss how this approach works better (or not) compared to the end-to-end optimization where discriminative training would be used with the exact same computational power used.

1. The authors need to also show the extra computational load that the VoiceFixer method induces compared to the gain that it delivers. Specifically, I have the following suggestions: The authors propose a complex architecture that might severely enlarge the actual memory footprint of a forward pass of the network (or backward) mainly because of the intermediate activations tensors. Moreover, the  authors should report the number of trainable parameters, the number of FLOPs as well as the actual memory requirements, time consumed during training on GPU / inference on CPU. Most importantly, this comparison could also be used to adress the concern above that would show that VoiceFixer helps even against the same computational load.

2. The training algorithm as described in the Appendix (Algorithm 1) could be utilized using any time-domain state of the art network. For example, I do not see the reason of why not directly using a state-of-the-art speech denoising network [3, 4, 5] and train it for all the 4 tasks using Algorithm 1 and simply put a few extra upsampling layers for the super-resolution audio task. Also, I would expect that the model comparisons, especially for tasks such as denoising, which have open-sourced benchmark datasets, to be conducted using datasets like the DNS-Challenge [6] (but this is more optional compared to the main point of direct comparison with current state-of-the-art models).

3. Although the authors provided a few sample spectrograms in the Appendix, it is really hard to understand from an image how the actual speech sounds. Since the MOS have been reported and generative setups have been employed it is really important to include a few random reconstructed speech files to assess the performance and let the reviewers actually hear the results of the method and compare.

4. The authors have identified that because of the use of generative setups it might be the case that point-wise metrics such as SISNR do not capture how realistic a generated speech file sounds. Although this is true in general, this decline in performance is often caused by time-domain shifts that can be easily fixed with the appropriate padding (e.g. the metric wrt log MSE on the spectrogram domain does not display a similar behavior, probably because the computation window is wider).

5. How the authors explain that for the super-resolution task (e.g. Figure 6e), the mean MOS for a few models are better than the high-resolution target itself and also worse than the unprocessed file? This is extremely odd and is not at all consistent with Figure 6b.

**Minor**
- There are probably more typos but here are a few that I have found:

    - we design a similar metrics → we design similar metrics
    - The experiment code can conduct evaluations → The code can be used to conduct the evaluations
    - Sections  4.1 and 4.4, include “:” after each bold word

- Last Paragraph of Section D3 Appendix:

    - all these examples prove the effectiveness → all these examples showcase the effectiveness
    - which proves the advantages → which displays the advantages
    - Last but not least, despite the abnormal harmonic structure in the low-frequency part in Figure 12g, our proposed model can still repair it into a normal distribution, → the phrasing repairing something into a normal distribution is both wrong and misleading.

- What is the reason where some vectors are denoted with bold and others not? (E.g. equation 29)

[1] Adam Polyak, Lior Wolf, Yossi Adi, Ori Kabeli, and Yaniv Taigman. High fidelity speech regeneration with application to speech enhancement. In Proceedings of the IEEE Conference on Acoustics, Speech, and Signal Processing, pp. 7143–7147, 2021.

[2] Efthymios Tzinis, Shrikant Venkataramani, Zhepei Wang, Cem Subakan, Paris Smaragdis. “Two-step sound source separation: Training on learned latent targets.” In ICASSP 2020-2020 IEEE International Conference on Acoustics, Speech and Signal Processing (ICASSP) 2020 May 4 (pp. 31-35).

[3] Umut Isik, Ritwik Giri, Neerad Phansalkar, Jean-Marc Valin, Karim Helwani, and Arvindh Krishnaswamy, “Poconet: Better speech enhancement with frequency-positional embeddings, semi-supervised conversational data, and biased loss,” in Proc. Interspeech, 2020, pp. 2487–2491.

[4] Ashutosh Pandey and DeLiang Wang, “Dense cnn with selfattention for time-domain speech enhancement,” IEEE/ACM Transactions on Audio, Speech, and Language Processing, vol. 29, pp. 1270–1279, 2021.

[5] Défossez, A., Synnaeve, G., Adi, Y. (2020) Real Time Speech Enhancement in the Waveform Domain. Proc. Interspeech 2020, pp. 3291-3295.

[6] Chandan KA Reddy et al., “The interspeech 2020 deep noise suppression challenge: Datasets, subjective testing framework, and challenge results,” in Proc. Interspeech, 2020.

**Summary Of The Paper:**

This paper introduces a unified view of several speech restoration problems including denoising, decliping, dereverberation and audio super-resolution. In order to solve the problem of the general speech restoration task, the authors propose a U-Net architecture which is trained on all of these tasks simultaneously during training time. The authors conduct extensive experiments for the general speech restoration task as well as the individual tasks where they compare the proposed models with more specialized models bounded to each distortion. The experimental results show that the proposed VoiceFixer combination of the model and the analysis-synthesis procedure are capable of effectively removing the speech distortions and in some cases outperform previous approaches in the literature.

**Summary Of The Review:**

Overall, this is a well written paper which discusses an interesting idea for addressing the problem of removing speech distortions in a more general way with strong results. The paper discusses many experimental design choices that researchers would need to replicate the results, which is a strong plus. It is also important that the authors have conducted subjective evaluation experiments to show how their algorithm performs. I think that the paper could have a great potential in the field, however, in its current form, the paper has several limitations which downplay its true potential. Specifically, showing how the proposed method is different from ones presented in the literature as well as why it is more important than simply employing more computational power. Moreoever, there are a few misconceptions in the experimental section that authors need to address before the paper is ready for a publication.

---

> ### Author Response · Authors · 2021-11-20
> **Thank you for the suggestions.**
>
> Thanks for the detailed and concise suggestions on my work. I really appreciate it. I've modified the minor typos and would liked to make the following clarifications.
>
> > 1. Although the authors provided a few sample spectrograms in the Appendix, it is really hard to understand from an image how the actual speech sounds. Since the MOS have been reported and generative setups have been employed it is really important to include a few random reconstructed speech files to assess the performance and let the reviewers actually hear the results of the method and compare.
>
> As was mentioned in the footnote of the first page, we have a demo page for readers:
>
> https://anonymous20211004.github.io/demo-vf/
>
> > 4. The authors have identified that because of the use of generative setups it might be the case that point-wise metrics such as SISNR do not capture how realistic a generated speech file sounds. Although this is true in general, this decline in performance is often caused by time-domain shifts that can be easily fixed with the appropriate padding (e.g. the metric wrt log MSE on the spectrogram domain does not display a similar behavior, probably because the computation window is wider).
>
> Yes, we did try with padding or searching optimal alignments (usually is a 10-50 samples shift), which still cannot get a normal result on time-domain metrics. We believe that's because our restriction for alignment during training is loose. Our vocoder uses a lot of discriminative losses and coarse-grain loss functions, which do not need the exact alignment to converge and produce a low loss value.
>
> > 5. How the authors explain that for the super-resolution task (e.g. Figure 6e), the mean MOS for a few models are better than the high-resolution target itself and also worse than the unprocessed file? This is extremely odd and is not at all consistent with Figure 6b.
>
> Figure 6e indicates, from the listener point of view, 24k sample rate and 44.1k sample rate speech is hard to distinguish, which is one of the reasons for the close scores between *target* and *unprocessed*. The result of SEANet being better than target seems pretty odd, but we observed SEANet's output has more energy in the higher frequency comparing with the target, which could potentially make the speech sound more clearer (might be preferred by the listener). In the 8kHz (Figure 6b) case, where the unprocessed file has a larger gap with the target, the performance gap between different models becomes more evident than 24k setting.

---

> > ### Comment · Reviewer_Q5gX · 2021-11-22
> > **Response to the Authors**
> >
> > I would like to thank the author (it is evident that there is a single author from the responses and the audio sample from their own voice in the website) for their responses and their effort to answer some of my concerns. Please also keep in mind that the usage of `I` and `my`  in the responses and the usage of personal author's recordings raise concerns about the anonymity of the identity of the author.
> >
> > >> R: Since the MOS have been reported and generative setups have been employed it is really important to include a few random reconstructed speech files to assess the performance...
> >
> > > A: As was mentioned in the footnote of the first page, we have a demo page for readers: https://anonymous20211004.github.io/demo-vf/
> >
> > As I mentioned explicitly in my first comment, I wanted to assess the performance on *random* examples, since the current examples do not seem randomly sampled (samples from author's voice or at least it was not stated) and they are extremely limited (one or two per task).
> >
> > > Figure 6e indicates, from the listener point of view, 24k sample rate and 44.1k sample rate speech is hard to distinguish, which is one of the reasons for the close scores between target and unprocessed. he result of SEANet being better than target seems pretty odd, but we observed SEANet's output has more energy in the higher frequency comparing with the target, which could potentially make the speech sound more clearer (might be preferred by the listener). In the 8kHz (Figure 6b) case, where the unprocessed file has a larger gap with the target, the performance gap between different models becomes more evident than 24k setting.
> >
> > If this is the case that it is normal to have higher or almost the same MOS for the unprocessed file than the target, then what is the point of the task? I am not entirely convinced that the evaluation is correct and the response did not alleviate my concern.
> >
> > I would be grateful if the reviewer could also comment on my other two concerns from my initial review.

---

### Official Review · Reviewer_UVJ3 · 2021-11-02

**Correctness:** 3
**Technical Novelty And Significance:** 2
**Empirical Novelty And Significance:** 3
**Recommendation:** 6
**Confidence:** 4

**Main Review:**

In this paper, the authors propose a single model-based general speech restoration (GSR) task for removing multiple distortions simultaneously. They also propose a two-stage-based generative framework called VoiceFixer to address the general speech restoration task. To develop and evaluate these models, they performed extensive experiments in the areas of speech restoration such as denoising, dereverberation, super-resolution, and declipping.
Their proposed single model-based GSR approach employing ResUNet provided improved results over the well-known existing single-task speech restoration techniques. VoiceFixer even surpasses their single model-based GSR approach. It is meaningful and impressive that the two proposed single-model-based approaches outperform the well-known existing single-task speech restoration techniques in almost every speech restoration tasks described in the paper.
The authors report that VoiceFixer outperforms the GSR approach in the experimental results. However, it is unclear which makes VoiceFixer superior to the GSR approach. The author needs to clarify this issue.
A weak point may be that VoiceFixer restores speech as a synthesis form by employing a neural vocoder which inevitably entails some degree of distortion when compared with real speech. Therefore, when the raw speech has relatively small amount of distortion, the restored speech is still vocoder-decoded speech which may be inferior to the uncorrupted clean speech. In this case, the merit of VoiceFixer over the IFFT-based approach may be decreased.


**Summary Of The Paper:**

This paper proposes a general speech restoration (GSR) task that tries to remove multiple distortions in a single model. In addition, it also presents a generative framework called VoiceFixer consisting of analysis and synthesis stages to address the general speech restoration task. In VoiceFixer, the authors employ a ResNet for modeling the analysis stage and a TFGAN-based neural vocoder for synthesis stage. They report that their baseline GSR and VoiceFixer surpass the single speech restoration (SSR) models with more improved results by the latter. Their idea was well described and the experiments are systematical and extensive. The results are consistent and clear.
The contribution of this paper is to incorporate a variety of speech restoration tasks including speech denoising, super-resolution, dereverberation, declipping, etc. in a single unified task called GSR. Another is the proposal of a well-performing generative speech restoration framework called VoiceFixer.


**Summary Of The Review:**

The reviewed paper proposed two new approaches for speech restoration. The first approach is a single-model-based generative speech restoration technique that can cope with a variety of speech restoration tasks simultaneously. The other is a generative speech restoration frame called VoiceFixer which provides enhanced performance over the well-known existing techniques as well as the proposed GSR approach.
In spite of some unclear and weak points mentioned above, the reviewer thinks the two proposed approaches highly and believes that they could impact positively on the related society.

---

> ### Author Response · Authors · 2021-11-20
> **Thank your for the comments and suggestions!**
>
> I appreciate your comments and suggestions. That's invaluable. And sure, some distortions may be introduced by the vocoder. But in general, the distortion of a well-trained vocoder is not easily perceptible. Moreover, I believe this problem is even less frequent in the low sample rate of single-speaker scenarios.

---

### Official Review · Reviewer_Hsxt · 2021-11-06

**Correctness:** 3
**Technical Novelty And Significance:** 2
**Empirical Novelty And Significance:** 2
**Recommendation:** 3
**Confidence:** 5

**Main Review:**

1. The paper claims that most speech restoration works are driven by a single task. I do not completely agree with this statement. There are plenty of works which have explored speech denoising and dereverberations together, conventional signal processing based methods as well as neural network based methods [R1, R3, R4, R5]. Some works such as HiFi-GAN also consider other types of distortions such as equalization distortions.


2. Overall, I find very little to no novelty in the paper with respect to the proposed method. Two stage approaches like the one proposed in the paper have been used before. In fact, the prior work [R2, R6] takes more or less the exact same approach for speech enhancement -- reconstruct acoustic features and then use vocoder.


3. Moreover, the paper does not propose any novel method for each stage. The first stage (analysis stage) uses a pre-existing architecture with well-known L1 based reconstruction loss. Similarly, the second stage also uses a prior vocoder (TFGAN) without any modifications.


4. On the empirical front also, the paper leaves out quite a few desirable and expected experiments. Experiments are done using only the VCTK dataset. VCTK and VCTK-demand are both relatively small datasets and do not provide the best picture of the performance. Moreover, one would expect comparisons with latest state of the art approaches for denoising (enhancement) and dereverberations. There are quite a lot of works in these areas and it is expected to compare with reasonable state of the art methods.


5. It would have been interesting if the paper showed results using different methods for mel-restoration (they could use a variety of prior works here) and different vocoders in stage 2. Considering that the two stage approach and the methods used in each stage have been done before, this would have helped establish the significance of different approaches for analysis and the synthesis stages. The significance of each stage are not properly decoupled as well. Can we keep the same vocoder but improve the overall system somehow by improving analysis stage ? Similarly, the other way round as well.


6. It's not clear to me if Denoise-UNet, Dereverb-UNet etc are single stage or 2 stage approaches. Results tables mention them as a one-stage model. How are restored speeches obtained ? Are they directly predicting clean speech or similar to the two stage predict restored mel and then uses vocoder for synthesis. In the latter case, these are also two stage approaches ??


7. The paper claims to be solving general speech restoration. This could have been a focus of the paper. Showing how training with all distortions helps in improving restoration in case of single as well as multiple distortions. However, this is not emphasized and clear through experiments and discussions.


8. While such a two stage approach makes sense, the connection between human hearing and the proposed approach is rather weak. I am not sure if it is a good idea to make that connection here.


9. Except for minor language issues here and there, the paper is clearly written. The authors have also provided demos and codes through anonymous github links.


10.  What happens if the two stages are trained jointly -- in an end to end fashion ? Is it better or worse than training separately ?


11. Some analysis w.r.t level of distortions (e.g SNR level for additive noise, RT60 for reverberations, amount of clipping etc.) would have helped better understand the performance.


R1: Joint Dereverberation And Noise Reduction Using Beamforming
And A Single-channel Speech Enhancement Scheme, Cauchi et. al., 2014


R2: Speaker Independence Of Neural Vocoders And Their Effect On Parametric Resynthesis Speech Enhancement, Maiti and Mandel, 2019.

R3: Denoising-and-dereverberation Hierarchical Neural Vocoder For Robust Waveform Generation, Ai et. al., 2020

R4: A Unified Convolutional Beamformer For Simultaneous Denoising And Dereverberation, Nakatani et. al., 2019.

R5: HiFi-GAN: High-Fidelity Denoising and Dereverberation Based on Speech Deep Features in Adversarial Networks, Su et. al., 2019

R6: Speech Enhancement Using Speech Synthesis Techniques, Maiti, 2021


**Summary Of The Paper:**

The paper proposes an approach called VoiceFixer which is aimed at restoring degraded speech signals. The paper considers a variety of speech degradations - additive noise, reverberations, clipping and limited bandwidth. The paper describes a two stage approach in which the first stage aims to produce restored mel-spectrogram and then a vocoder is used to synthesize the  speech from the restored mel-spectrogram. Experiments are done using the VCTK dataset and experiments are done using single distortions as well as combinations of all 4 distortions.

**Summary Of The Review:**

The novelty of the paper in terms of the methods is not much. The analysis and synthesis approach as well as the method for each stage are not particularly novel. Several prior works have used such methods. The experimental results and analyses are also not strong and does not provide interesting insights. Overall, the paper lacks strength on both fronts in the current form.

---

> ### Author Response · Authors · 2021-11-20
> **Thanks for the suggestions!**
>
> Thanks for your detailed feedback and suggestions. I appreciate the points you made but some of them may need further clarifications.
>
> > 1. The paper claims that most speech restoration works are driven by a single task. I do not completely agree with this statement. There are plenty of works which have explored speech denoising and dereverberations together, conventional signal processing based methods as well as neural network based methods [R1, R3, R4, R5]. Some works such as HiFi-GAN also consider other types of distortions such as equalization distortions.
>
> I suppose most restoration works focus on a single task because tasks like speech enhancement are well defined for a long time and receive a lot of attentions. While joint performing tasks are still a less heated research area and usually not as 'general' as our proposed model, which also handles two more challenging tasks, super-resolution and clipping. And four types of distortions are not the limit of our system. Tasks like equalization restoration can be added easily to our model by modifying the distortion simulation process.
>
> > 3. Moreover, the paper does not propose any novel method for each stage. The first stage (analysis stage) uses a pre-existing architecture with well-known L1 based reconstruction loss. Similarly, the second stage also uses a prior vocoder (TFGAN) without any modifications.
>
> Because we think the simplest is the best. We tried many other strategies like adversarial training or fine-tuning speakers. But we found the simple architecture we finally show is the best to demonstrate our core idea.
>
> > 4. On the empirical front also, the paper leaves out quite a few desirable and expected experiments. Experiments are done using only the VCTK dataset. VCTK and VCTK-demand are both relatively small datasets and do not provide the best picture of the performance. Moreover, one would expect comparisons with latest state of the art approaches for denoising (enhancement) and dereverberations. There are quite a lot of works in these areas and it is expected to compare with reasonable state of the art methods.
>
> We only use VCTK and VCTK-demand because we'd like to make our work easy to reproduce. Nevertheless, I think VCTK is already quite enough for experiment with 44 hours of data and 109 speakers and is widely used in super-resolution and text-to-speech tasks such as R1, R2, and R3.
>
> > 6. It's not clear to me if Denoise-UNet, Dereverb-UNet etc are single stage or 2 stage approaches. Results tables mention them as a one-stage model. How are restored speeches obtained ? Are they directly predicting clean speech or similar to the two stage predict restored mel and then uses vocoder for synthesis. In the latter case, these are also two stage approaches ??
>
> The restoration process of the one-stage UNet model is described in 3.1. We keep this one-stage restoration method as close to the methodology of the analysis module in 3.2.1 as possible for better comparisons.
>
> > 7. The paper claims to be solving general speech restoration. This could have been a focus of the paper. Showing how training with all distortions helps in improving restoration in case of single as well as multiple distortions. However, this is not emphasized and clear through experiments and discussions.
>
> We put a lot of emphasis on the exploration of general speech restoration. We made a GSR test set and perform both subjective and objective evaluations on it. We also described the process of data simulation in detail in Appendix C.2.
>
> > 11. Some analysis w.r.t level of distortions (e.g SNR level for additive noise, RT60 for reverberations, amount of clipping etc.) would have helped better understand the performance.
>
> We think the detailed numbers would distract readers. So we described the settings of different kinds of distortions in Appendix C.1 and Appendix C.2. But, sure, the level of distortions can be helpful for the reader to understand.
>
>
>
> R1. Time-frequency Network for Audio Super-resolution, Lim Teck Yian, et al. 2018
>
> R2. Time-frequency Loss for CNN Based Speech Super-Resolution, Heming Wang, and DeLiang Wang, 2020
>
> R3. NU-Wave: A Diffusion Probabilistic Model for Neural Audio Upsampling, Lee Junhyeok, and Seungu Han, 2021

---

> > ### Comment · Reviewer_Hsxt · 2021-11-28
> > **Response to Authors Rebuttal**
> >
> > Thanks for responding to the reviews. My key concerns regarding this paper remains and I also agree with the comments from other reviewers (several of which have not been addressed in the rebuttal). I am keeping the score as is.

---

### Decision · Program_Chairs · 2022-01-20

**Decision:**

Reject

**Comment:**

PAPER: This paper addresses the problem of learning methods for general speech restoration which generalizes across at least 4 tasks (additive noise, room reverberation, low-resolution and clipping distortion).  The proposed approach is based on a two-stage process, which includes both analysis and synthesis stages.
DISCUSSION: The reviewers wrote very detailed reviews which ask some important questions and point to some potential issues. The authors responded to all reviews, but only addressed a subset of the issues and questions mentioned by the reviewers. Novelty and comparison with previous approaches was one of the issues mentioned by reviewers.
SUMMARY: While reviewers are supportive of this line of research, reviewers were also concerned with the novelty of the proposed approach and details of the experiments. In its current form, the paper may not be ready for publication.